# Protein phosphatase 1 activity controls a balance between collective and single cell modes of migration

Yujun Chen[1], Nirupama Kotian[1], George Aranjuez[2†], Lin Chen[3], C Luke Messer[1], Ashley Burtscher[2], Ketki Sawant[1], Damien Ramel[3‡], Xiaobo Wang[3], Jocelyn A McDonald[1]*

[1]Division of Biology, Kansas State University, Manhattan, United States; [2]Lerner Research Institute, Cleveland Clinic, Cleveland, United States; [3]LBCMCP, Centre de Biologie Intégrative (CBI), Université de Toulouse, CNRS, UPS, Toulouse, France

*For correspondence: jmcdona@ksu.edu

Present address: †Division of Immunity and Pathogenesis, Burnett School of Biomedical, Sciences, University of Central Florida College of Medicine, Orlando, United States; ‡Institute of Metabolic and Cardiovascular Diseases (I2MC), Université de Toulouse, Institut National de la Santé et de la Recherche Médicale (INSERM) UMR1048, Toulouse, France

**Competing interests:** The authors declare that no competing interests exist.

**Abstract** Collective cell migration is central to many developmental and pathological processes. However, the mechanisms that keep cell collectives together and coordinate movement of multiple cells are poorly understood. Using the *Drosophila* border cell migration model, we find that Protein phosphatase 1 (Pp1) activity controls collective cell cohesion and migration. Inhibition of Pp1 causes border cells to round up, dissociate, and move as single cells with altered motility. We present evidence that Pp1 promotes proper levels of cadherin-catenin complex proteins at cell-cell junctions within the cluster to keep border cells together. Pp1 further restricts actomyosin contractility to the cluster periphery rather than at individual internal border cell contacts. We show that the myosin phosphatase Pp1 complex, which inhibits non-muscle myosin-II (Myo-II) activity, coordinates border cell shape and cluster cohesion. Given the high conservation of Pp1 complexes, this study identifies Pp1 as a major regulator of collective versus single cell migration.

## Introduction

Cells that migrate as collectives help establish and organize many tissues and organs in the embryo, yet also promote tumor invasion, dissemination and metastasis (*Friedl et al., 2012*; *Friedl and Gilmour, 2009*; *Wang et al., 2016*; *Cheung and Ewald, 2016*; *Scarpa and Mayor, 2016*). A wide variety of cells undergo collective cell migration during development, ranging from neural crest cells in *Xenopus*, the zebrafish lateral line primordium, and branching mammary glands (*Friedl and Gilmour, 2009*; *Scarpa and Mayor, 2016*; *Huebner et al., 2016*; *Shellard and Mayor, 2019*), among many other examples. Despite the apparent diversity in collectively migrating cell types, there is remarkable conservation of the cellular and molecular mechanisms that underlie group cell movements. In particular, migrating collectives require fine-tuned organization and cell coordination to move effectively as a unified group. Similar to individually migrating cells, collectively migrating cells display a front-rear polarity, but this polarity is often organized at the group level (*Mayor and Etienne-Manneville, 2016*). Leader cells at the front extend characteristic protrusions that help collectives navigate tissues. Mechanical cell coupling and biochemical signals then reinforce collective polarity by actively repressing protrusions from follower cells and by maintaining lead cell protrusions that pull the group forward (*Mayor and Etienne-Manneville, 2016*; *Friedl and Mayor, 2017*). Importantly, cell-cell adhesions keep collectives together by maintaining strong but flexible connections between cells. Moreover, many cell collectives exhibit a 'supracellular' organization of the cytoskeleton at the outer perimeter of the entire cell group that serves to further coordinate multicellular movement (*Shellard and Mayor, 2019*; *Shellard et al., 2018*; *Hidalgo-Carcedo et al., 2011*; *Reffay et al., 2014*). Despite progress in understanding how single cells become polarized and motile, less is

known about the mechanisms that control the global organization, cohesion, and coordination of cells in migrating collectives.

*Drosophila* border cells are a genetically tractable and relatively simple model well-suited to investigate how cell collectives undergo polarized and cooperative migration within a developing tissue (*Montell et al., 2012*; *Saadin and Starz-Gaiano, 2016*). The *Drosophila* ovary is composed of strings of ovarioles made up of developing egg chambers, the functional unit of the *Drosophila* ovary. During late oogenesis, four to eight follicle cells are specified at the anterior end of the egg chamber to become migratory border cells. The border cells then surround a specialized pair of follicle cells, the polar cells, and delaminate as a multicellular cluster from the follicular epithelium. Subsequently, the border cell cluster undergoes a stereotyped collective migration, moving between 15 large germline-derived nurse cells to eventually reach the oocyte at the posterior end of the egg chamber (*Figure 1A–F*). Throughout migration, individual border cells maintain contacts with each other and with the central polar cells so that all cells move as a single cohesive unit (*Llense and Martín-Blanco, 2008*; *Cai et al., 2014*). A leader cell at the front extends a migratory protrusion whereas protrusions are suppressed in trailing follower cells (*Prasad and Montell, 2007*; *Bianco et al., 2007*; *Poukkula et al., 2011*). As with other collectives, polarization of the border cell cluster is critical for the ability to move together and in the correct direction, in this case towards the oocyte (*Figure 1A–F*; *Prasad and Montell, 2007*; *Bianco et al., 2007*).

Polarization of the border cell cluster begins when two receptor tyrosine kinases (RTKs) expressed by border cells, PDGF- and VEGF-receptor related (PVR) and Epidermal Growth Factor Receptor (EGFR), respond to multiple growth factors secreted from the oocyte (*Duchek et al., 2001*; *McDonald et al., 2006*). Signaling through PVR/EGFR increases activation of the small GTPase Rac, triggering F-actin polymerization and formation of a major protrusion in the lead border cell (*Prasad and Montell, 2007*; *Poukkula et al., 2011*; *Duchek et al., 2001*; *Wang et al., 2010*). E-Cadherin-based adhesion to the nurse cell substrate stabilizes this lead cell protrusion via a feedback loop with Rac (*Cai et al., 2014*). Furthermore, the endocytic protein Rab11 and the actin-binding protein Moesin mediate communication between border cells to restrict Rac activation to the lead cell (*Ramel et al., 2013*). Mechanical coupling of border cells through E-Cadherin suppresses protrusions in follower cells, both at cluster exterior surfaces but also between border cells and at contacts with polar cells (*Montell et al., 2012*; *Cai et al., 2014*). E-Cadherin also maintains border cell attachment to the central polar cells. F-actin and non-muscle myosin II (Myo-II) are enriched at the outer edges of the border cell cluster (*Aranjuez et al., 2016*; *Lucas et al., 2013*; *Combedazou et al., 2017*). Such 'inside-outside' polarity contributes to the overall cluster shape, cell-cell organization, and coordinated motility of all border cells (*Montell et al., 2012*). While progress has been made in understanding the establishment of front-rear polarity, much less is known about how individual border cell behaviors are fine-tuned and adjusted to produce coordinated and cooperative movement of the cluster as an entire unit.

In the current study we made the unexpected discovery that Protein phosphatase 1 (Pp1) activity coordinates the collective behavior of individual border cells. Dynamic cycles of protein phosphorylation and dephosphorylation precisely control many signaling, adhesion and cytoskeletal pathways required for cell migration (*Larsen et al., 2003*). Serine-threonine kinases, such as Par-1, Jun kinase (JNK), and the p21-activated kinase Pak3, as well as phosphorylated substrate proteins such as the Myo-II regulatory light chain (MRLC; *Drosophila* Spaghetti squash, Sqh) and Moesin regulate different aspects of border cell migration (*Llense and Martín-Blanco, 2008*; *Ramel et al., 2013*; *Majumder et al., 2012*; *Felix et al., 2015*). In contrast, the serine-threonine phosphatases that counteract these and other kinases and phosphorylation events have not been extensively studied, either in border cells or in other cell collectives. Pp1 is a highly conserved and ubiquitous serine-threonine phosphatase found in all eukaryotic cells (*Lin et al., 1999*; *Verbinnen et al., 2017*). Pp1 can directly dephosphorylate substrates in vitro, but specificity for phosphorylated substrates in vivo is generally conferred by a large number of regulatory subunits (also called Pp1-interacting proteins [PIPs]). These regulatory subunits form functional Pp1 complexes through binding to the Pp1 catalytic (Pp1c) subunits and mediate the recruitment of, or increase the affinity for, particular substrates (*Verbinnen et al., 2017*; *Heroes et al., 2013*). Thus, despite the potential for pleiotropy, Pp1 complexes have specific and precise cellular functions in vivo, that range from regulation of protein synthesis, cell division and apoptosis to individual cell migration (*Ceulemans and Bollen, 2004*; *Ferreira et al., 2019*).

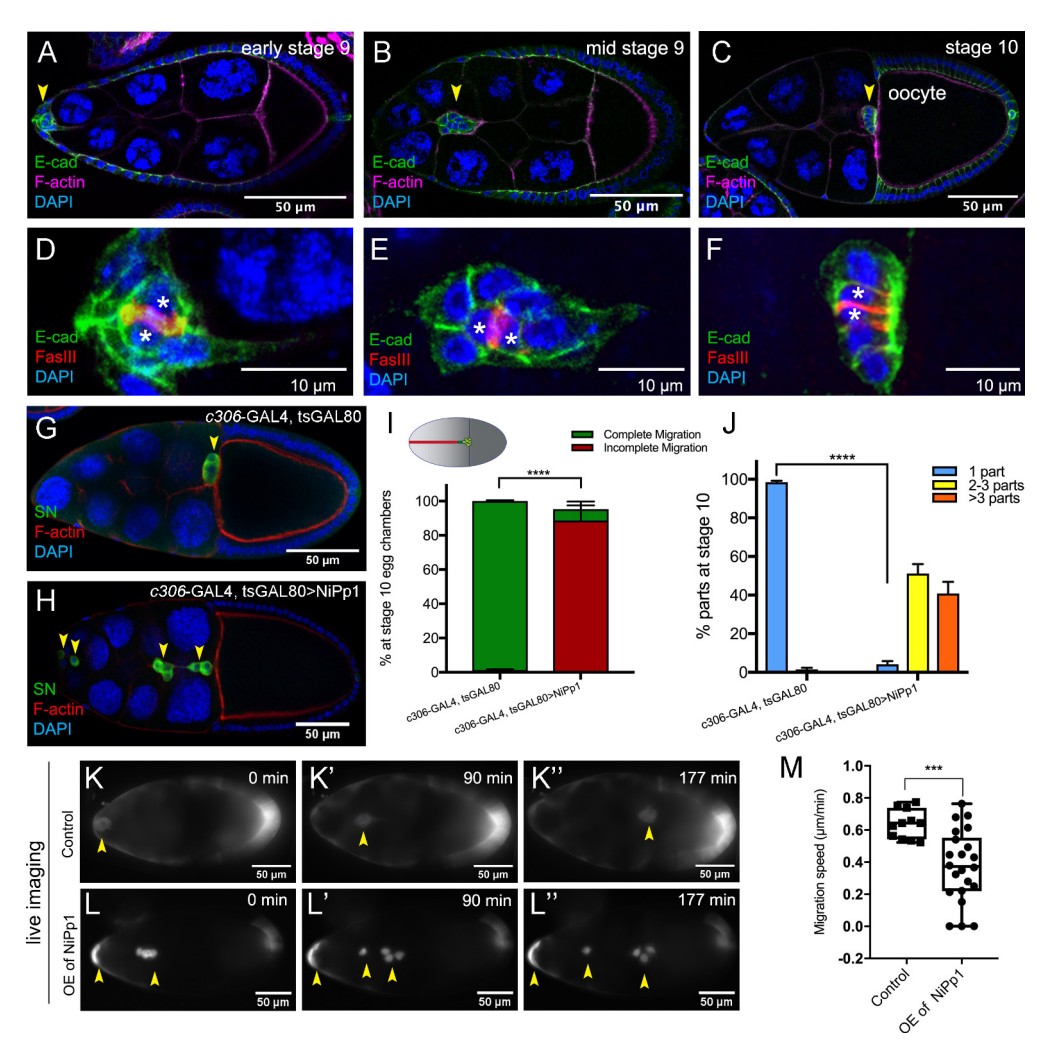

**Figure 1.** NiPp1 expression causes the border cell cluster to fall apart and disrupts migration. (**A–F**) Wild-type border cell migration during oogenesis stages 9 and 10. (**A–C**) Egg chambers at the indicated stages labeled with E-Cadherin (E-Cad; green), F-actin (magenta) and DAPI (blue). Arrowheads indicate the border cell cluster. (**D–F**) Magnified views of the same border cell cluster from (**A–C**), showing FasIII (red) in the polar cells, E-Cad and DAPI. The border cell cluster is composed of two polar cells (marked by asterisks) in the center and four to eight outer border cells that are tightly connected with each other as indicated by E-Cad staining. (**G, H**) Egg chambers labeled with Singed (SN; green) to detect border cells (arrowheads), phalloidin to detect F-actin (red), and DAPI to detect nuclei (blue). Control border cells (**G**) reach the oocyte as a single cluster, whereas NiPp1-expressing border cells (**H**) dissociate from the cluster into small groups, with only a few reaching the oocyte. (**I**) Quantification of border cell cluster migration for matched control and NiPp1 overexpression, shown as the percentage that did not complete (red), or completed (green) their migration to the oocyte, as indicated in the egg chamber schematic. (**J**) Quantification of cluster cohesion, shown as the percentage of border cells found as a single unit (one part) or split into multiple parts (2–3 parts or >3 parts) in control versus NiPp1-expressing egg chambers. (**I, J**) Error bars represent SEM in three experiments, each trial assayed n ≥ 69 egg chambers (total n ≥ 221 egg chambers per genotype). \*\*\*p<0.001, \*\*\*\*p<0.0001, unpaired two-tailed *t* test. (**K–L''**) Frames from a control (*Video 1*; K–K'') and an NiPp1 overexpression (OE; *Video 2*; L–L'') time-lapse video showing movement of the border cell cluster over the course of 3 hr (time in minutes). Border cells (arrowheads) express UAS-mCherry-Jupiter, which labels cytoplasmic microtubules. (**M**) Measurement of border cell migration speed from control (n = 11 videos) and NiPp1 overexpression (n = 11 videos; 22 tracked border cell 'parts' videos, shown as a box-and-whiskers plot. The whiskers represent the minimum and maximum; the box extends from the 25th to the 75th percentiles and the line indicates the median. \*\*\*\*p<0.0001, unpaired two-tailed *t* test. In this and all subsequent

*Figure 1 continued on next page*

*Figure 1 continued*

figures, anterior is to the left and the scale bars indicate the image magnification. All genotypes are listed in *Table 2*.

The online version of this article includes the following figure supplement(s) for figure 1:

**Figure supplement 1.** Patterns of GAL4s expressed in border cells.
**Figure supplement 2.** Cell-specific phenotypes induced by NiPp1.
**Figure supplement 3.** NiPp1 does not greatly alter border cell specification or cell number per cluster.

We now show that Pp1 activity controls multiple collective behaviors of border cells, including timely delamination from the epithelium, collective polarization, cohesion, cell-cell coordination, and migration. Remarkably, Pp1-inhibited border cells round up, break off from the main group, and move as single cells or small groups but are generally unable to complete their migration. We determine that Pp1 controls the levels of E-Cadherin and β-Catenin, which are needed to retain border cells within a cohesive cluster. Additionally, Pp1 activity restricts F-actin and Myo-II enrichment to the outer edges of the cluster, maintaining a supracellular cytoskeletal ultrastructure and supporting polarized collective movement. Furthermore, a major Pp1 specific complex for Myo-II activity, myosin phosphatase, coordinates border cell shape and adherence of cells to the cluster. Our work thus identifies Pp1 activity, mediated through distinctive phosphatase complexes such as myosin phosphatase, as a critical molecular regulator of collective cell versus single cell behaviors in a developmentally migrating collective.

## Results

### NiPp1 blocks border cell collective movement and cohesion in vivo

To address the role of phosphatases in border cell migration, we carried out a small-scale genetic screen to inhibit selected serine-threonine phosphatases that are expressed during oogenesis using RNAi as well as a protein inhibitor that targets Pp1 catalytic subunits (*Table 1*; *Miskei et al., 2011*; *Bennett et al., 2003*). We drove expression of RNAi and the inhibitor using *c306*-GAL4, an early anterior follicle cell driver expressed at high levels in border cells and polar cells (*Figure 1—figure supplement 1A*). Inhibition of *Pp4-19C* (one RNAi line) and Pp1c, through overexpression of Nuclear inhibitor of Protein phosphatase 1 (NiPp1), significantly disrupted border cell migration (*Table 1*). NiPp1 is an endogenous protein that when overexpressed, effectively and specifically blocks Pp1 catalytic subunit activity in vivo (*Bennett et al., 2003*; *Winkler et al., 2015*; *Parker et al., 2002*; *Van Eynde et al., 1995*). Pp1 and associated complexes are important phosphatase regulators of many cellular processes. Moreover, females expressing NiPp1 driven by *c306*-GAL4 did not produce adult progeny when crossed to wild-type males, consistent with infertility and suggesting a role for Pp1 in normal oogenesis (*Figure 1—figure supplement 2A*). Here we focused on further elucidating the function of Pp1 in border cells.

We used two GAL4 drivers to assess phenotypes, *c306*-GAL4 to determine early broad function of Pp1 in border cells and polar cells and *slbo*-GAL4 for later more restricted function in just border cells (*Figure 1—figure supplement 1*). Expression of NiPp1 strongly disrupted both the ability of border cells to organize into a cohesive cluster and to migrate successfully (*Figure 1G–J*). Unlike control border cells, most NiPp1-expressing border cells failed to reach the oocyte by stage 10 (98%; *Figure 1I*). Importantly, NiPp1-expressing border cells were no longer found in one cohesive cluster. Instead, individual cells and smaller groups split off from the main cluster (*Figure 1H*). Whereas control border cells migrated as a single cohesive unit ('one part'), NiPp1-expressing border cells split into two to three (50%), or more (40%), parts (*Figure 1H,J*). Migration and cluster cohesion defects were observed when NiPp1 was expressed early in both border cells and the central polar cells (*c306*-GAL4; *Figure 1I,J*; *Figure 1—figure supplement 2B*) or later in just border cells (*slbo*-GAL4; *Figure 1—figure supplement 2C–G*). Polar cells, through JAK/STAT signaling, recruit border cells to form a migratory cluster, and anchor border cells to the cluster (*Cai et al., 2014*; *Ghiglione et al., 2002*; *Silver and Montell, 2001*). Therefore, we tested the function of Pp1 in polar cells. We observed no defects in cohesion or migration when NiPp1 was expressed only in polar cells (*upd*-GAL4; *Figure 1—figure supplement 2C,H–K*). Fragmentation of clusters, however,

**Table 1.** Summary of the PPP family screen.

Results of the targeted serine-threonine protein phosphatase RNAi screen.

| Gene symbol | Annotation symbol | RNAi line | Migration defect (c306-Gal4) | Expression level in ovary (modENCODE) |
|---|---|---|---|---|
| Pp2B-14D | CG9842 | BDSC:25929 | No | moderate |
|  |  | BDSC:40872 | No |  |
|  |  | VDRC:46873 | No |  |
| mts | CG7109 | BDSC:27723 | Pupal lethal | moderate |
|  |  | BDSC:38337 | No |  |
|  |  | BDSC:57034 | No |  |
|  |  | BDSC:60342 | No |  |
| Pp4-19C | CG32505 | BDSC:27726 | Pupal lethal | moderate |
|  |  | BDSC:38372 | No |  |
|  |  | BDSC:57823 | Pupal lethal |  |
|  |  | VDRC:25317 | Yes |  |
| CanA-14F | CG9819 | BDSC:38966 | No | moderate |
| PpD3 | CG8402 | BDSC:57307 | No | moderate |
| PpV | CG12217 | BDSC:57765 | No | moderate |
| NiPp1 | CG8980 | BDSC:23711 | Yes | moderate |
| CanA1 | CG1455 | BDSC:25850 | No | low |
| CG11597 | CG11597 | BDSC:57047 | No | very low |
|  |  | BDSC:61988 | No |  |
| rgdC | CG44746 | BDSC:60076 | No | very low |

was stronger when NiPp1 was driven by *c306*-GAL4 rather than *slbo*-GAL4 (compare *Figure 1J* to *Figure 1—figure supplement 2G*), possibly due to earlier and higher expression of c306-GAL4 (*Figure 1—figure supplement 1*; *Silver and Montell, 2001*). Although polar cells are normally located at the center of the border cell cluster and maintain overall cluster organization (*Cai et al., 2014*; *Niewiadomska et al., 1999*), individual NiPp1-expressing border cells could completely separate from polar cells as well as the other border cells (*Figure 1—figure supplement 2L–N*). Finally, NiPp1 border cells appeared rounder than normal, indicating that individual cell shapes were altered (see below). Together, these results demonstrate that NiPp1 expression in border cells, but not polar cells alone, disrupts collective migration, cluster organization and adhesion.

Because very few border cells reached the oocyte, we investigated whether NiPp1-expressing border cells were correctly specified and functional. We first examined the expression of the transcription factor Slbo, the fly C/EBP homolog, which is required for border cell specification in response to JAK/STAT signaling (*Silver and Montell, 2001*; *Montell et al., 1992*). NiPp1-expressing border cells generally expressed Slbo, similarly to control cells (*Figure 1—figure supplement 3A–B'*; 30/33 border cells expressed Slbo, n = 6 egg chambers). Proper specification through JAK/STAT signaling restricts the number of follicle cells that become migrating border cells (*Silver and Montell, 2001*; *Starz-Gaiano et al., 2008*). When NiPp1 expression was driven by *c306*-GAL4, the total number of cells in the cluster (border cells and polar cells) was slightly increased to a mean of seven NiPp1 cells compared to six control cells per cluster (*Figure 1—figure supplement 3C*; n = 27 egg chambers for each genotype). This modest increase in cells per cluster is far fewer than what is observed upon ectopic activation of JAK/STAT (*Silver and Montell, 2001*; *Starz-Gaiano et al., 2008*), suggesting that NiPp1 does not greatly impact the specification or recruitment of border cells. Thus, NiPp1 prevents properly specified border cells from staying together and completing migration.

## Live NiPp1 border cell clusters fall apart and move slowly

To determine where and when NiPp1-expressing border cells stopped migrating and dissociated from the cluster, we examined border cell clusters using live time-lapse imaging (*Prasad and Montell, 2007*; *Dai and Montell, 2016*). Both control and NiPp1 border cells delaminated from the surrounding epithelium and began their migration as a group (*Figure 1K–L''*; *Videos 1*, *2*, *3*, *4*). NiPp1 border cells separated into multiple sub-clusters or single cells at various points during migration, particularly after moving between the nurse cells (*Videos 2*, *3*, *4*). NiPp1 border cells typically migrated as small groups but also could arrange themselves into co-linear chains (*Video 3*). A few NiPp1 border cells reached the oocyte, although considerably later than control border cells. Indeed, NiPp1-expressing border cells migrated more slowly overall compared to control border cell clusters (~0.35 µm/min NiPp1 versus ~0.65 µm/min control; *Figure 1M*). Individual NiPp1 border cells also moved at variable speeds, with lagging border cells sometimes pushing ahead of the nominal leading cell (*Video 2*). Labeling with a cortical cell membrane marker, PLCδ-PH-EGFP (*slbo*-GAL4 >UAS-PLCδ-PH-EGFP), allowed us to determine that some NiPp1 border cells completely

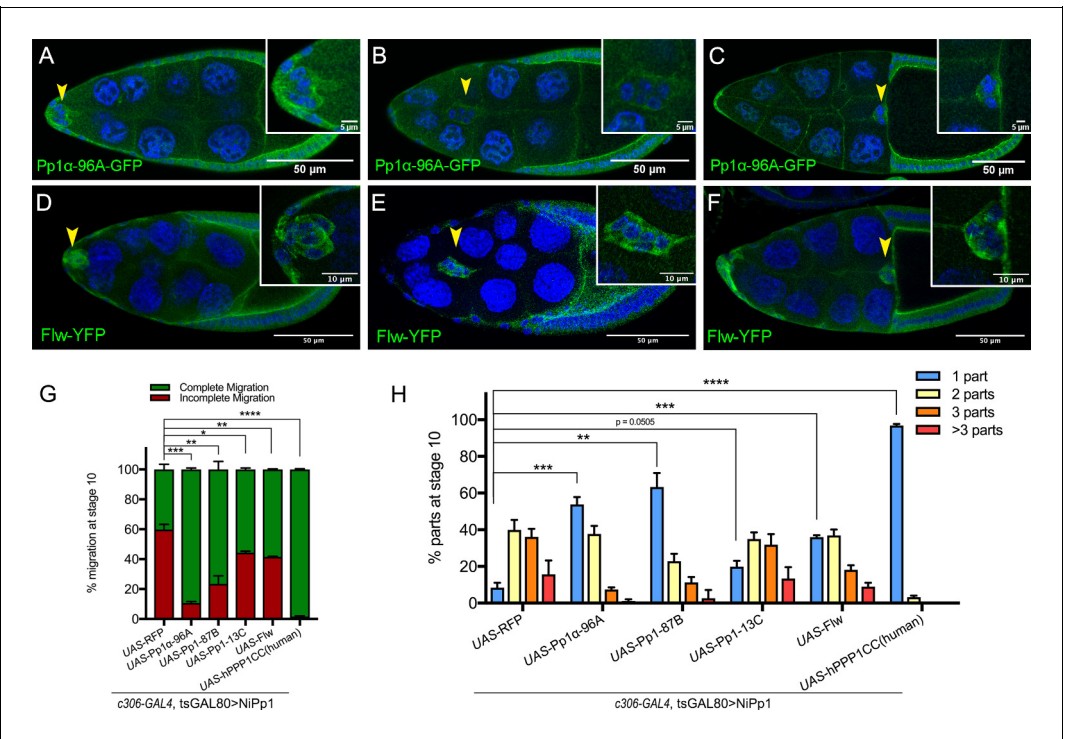

**Figure 2.** Pp1c expression in border cells and specificity of NiPp1 inhibition of Pp1c activity. (**A–F**) Stage 9 and 10 egg chambers showing the endogenous patterns of Pp1c subunits (green) in border cells (arrowheads), follicle cells, and the germline nurse cells and oocyte. DAPI (blue) labels nuclei. Insets, zoomed-in detail of border cells from the same egg chambers. (**A–C**) Pp1α−96A (green) expression, visualized by a GFP-tagged fly-TransgeneOme (fTRG) line. (**D–F**) Flw expression (green), visualized by a YFP-protein trap in the endogenous *flw* genetic locus. (**G, H**) Overexpression of *Pp1c* genes rescues the migration (**G**) and cluster cohesion (**H**) defects of NiPp1-expressing border cells. (**G**) Quantification of the migration distance at stage 10 for border cells in NiPp1-expressing egg chambers versus rescue by overexpression of the indicated *Pp1c* genes, shown as complete (green) and incomplete (red) border cell migration (see *Figure 1I* for egg chamber schematic). (**H**) Quantification of cluster cohesion at stage 10, shown as the percentage of border cells found as a single unit (one part) or split into multiple parts (two parts, three parts,>3 parts) in NiPp1-expressing egg chambers versus rescue by overexpression of the indicated *Pp1c* genes. (**G, H**) Error bars represent SEM in three experiments, each trial assayed n ≥ 44 egg chambers (total n ≥ 148 per genotype). *p<0.05, **p<0.01; ***p<0.001; ****p<0.0001, unpaired two-tailed *t* test. All genotypes are listed in *Table 2*.

The online version of this article includes the following figure supplement(s) for figure 2:

**Figure supplement 1.** Rescue of NiPp1 phenotypes by Pp1c genes.

**Figure supplement 2.** NiPp1 promotes nuclear localization of Pp1c subunits.

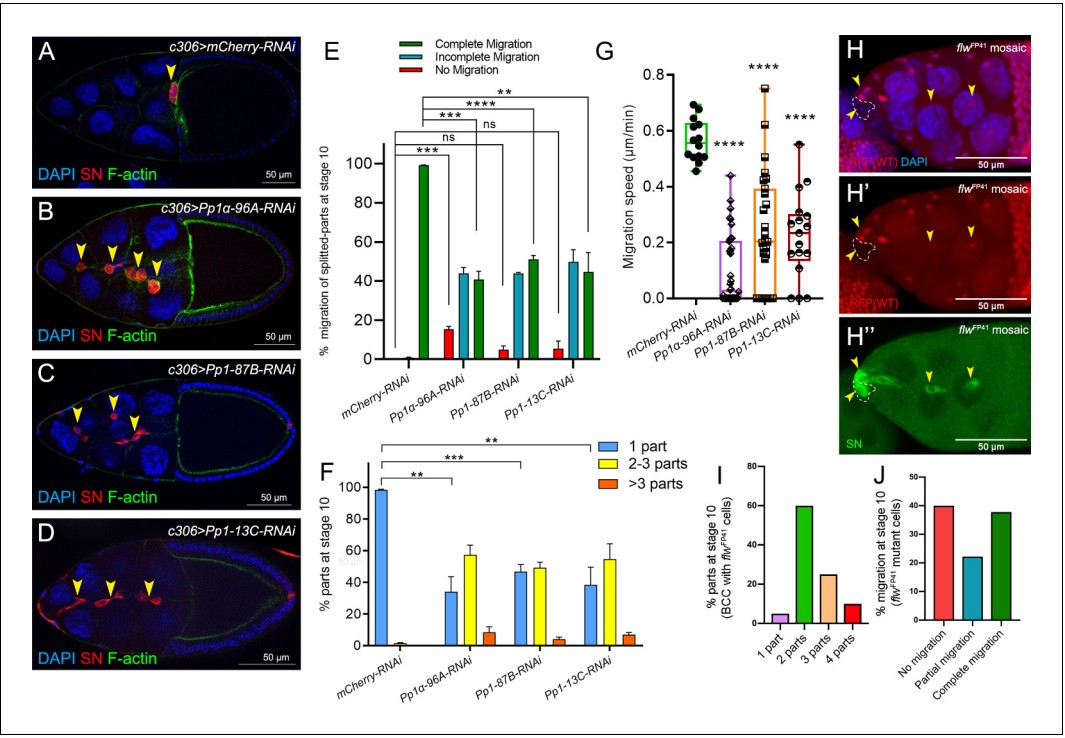

**Figure 3.** Pp1c genes are required for normal border cell migration and cluster cohesion. (A–F) Knockdown of *Pp1c* genes by RNAi disrupts border cell cluster migration and cohesion. (A–D) Stage 10 egg chambers expressing RNAi against the indicated genes were stained for SN (red) to label border cells (arrowheads), phalloidin to label F-actin (green) and DAPI to label nuclei (blue). (E) Quantification of border cell cluster migration for matched control and RNAi knockdown of the indicated *Pp1c* genes, shown as the percentage of egg chambers with complete (green), partial (blue), or no (red) border cell migration. (F) Quantification of cluster cohesion, shown as the percentage of border cells found as a single unit (one part) or split into multiple parts (2–3 parts or >3 parts) in control versus *Pp1c* RNAi egg chambers. (E, F) Error bars represent SEM in three experiments, each trial assayed n ≥ 58 (total n ≥ 229 per genotype). (G) Measurement of border cell migration speed in the indicated genotypes from individual videos of *Pp1c* RNAi border cells; n = 14 videos for control, n = 11 videos for *Pp1-87B-RNAi* (27 split parts were tracked), n = 12 videos for *Pp1-13C-RNAi* (17 split parts were tracked), n = 16 videos for *Pp1α-96A-RNAi* (38 split parts were tracked), box-and-whiskers plot (see *Figure 1* legend for details of plot). (E–G) *p<0.05, **p<0.01, ***p<0.001, ****p<0.0001, unpaired two-tailed *t* test. (H–J) *flw* mutant border cells split from the cluster and often fail to migrate. (H–H'') Representative image of a stage 10 egg chamber with *flw^FP41* mutant clones, marked by the loss of nuclear mRFP (dotted outline in H, H') and stained for SN (green in H'') to mark border cells (arrowheads) and DAPI (blue in H) to mark nuclei. (I, J) Quantification of *flw^FP41* mutant cluster cohesion (I) and migration (J) at stage 10; n = 20 egg chambers with *flw^FP41* clones were examined. (I) Quantification of cluster cohesion at stage 10, shown as the percentage of *flw^FP41* mosaic border cells found as a single unit (one part) or split into multiple parts (2, 3, or four parts). (J) Quantification of the migration distance at stage 10 for *flw^FP41* mosaic mutant border cells, shown as complete (green), partial (blue), or incomplete (red) border cell migration. All genotypes are listed in *Table 2*.

The online version of this article includes the following figure supplement(s) for figure 3:

**Figure supplement 1.** Delamination and migration defects caused by loss of *Pp1c*.

disrupted their cell-cell contacts, whereas other border cells remained in contact (*Video 5*). Finally, single border cells that broke off from the cluster were frequently left behind and stopped moving forward, appearing to get 'stuck' between nurse cells (*Videos 2*, *3*, *4*). Taken together, these data show that NiPp1 disrupts the ability of border cells to maintain a collective mode of migration, and leads to border cells now moving as single cells or small groups with slower speed that typically fail to reach the oocyte.

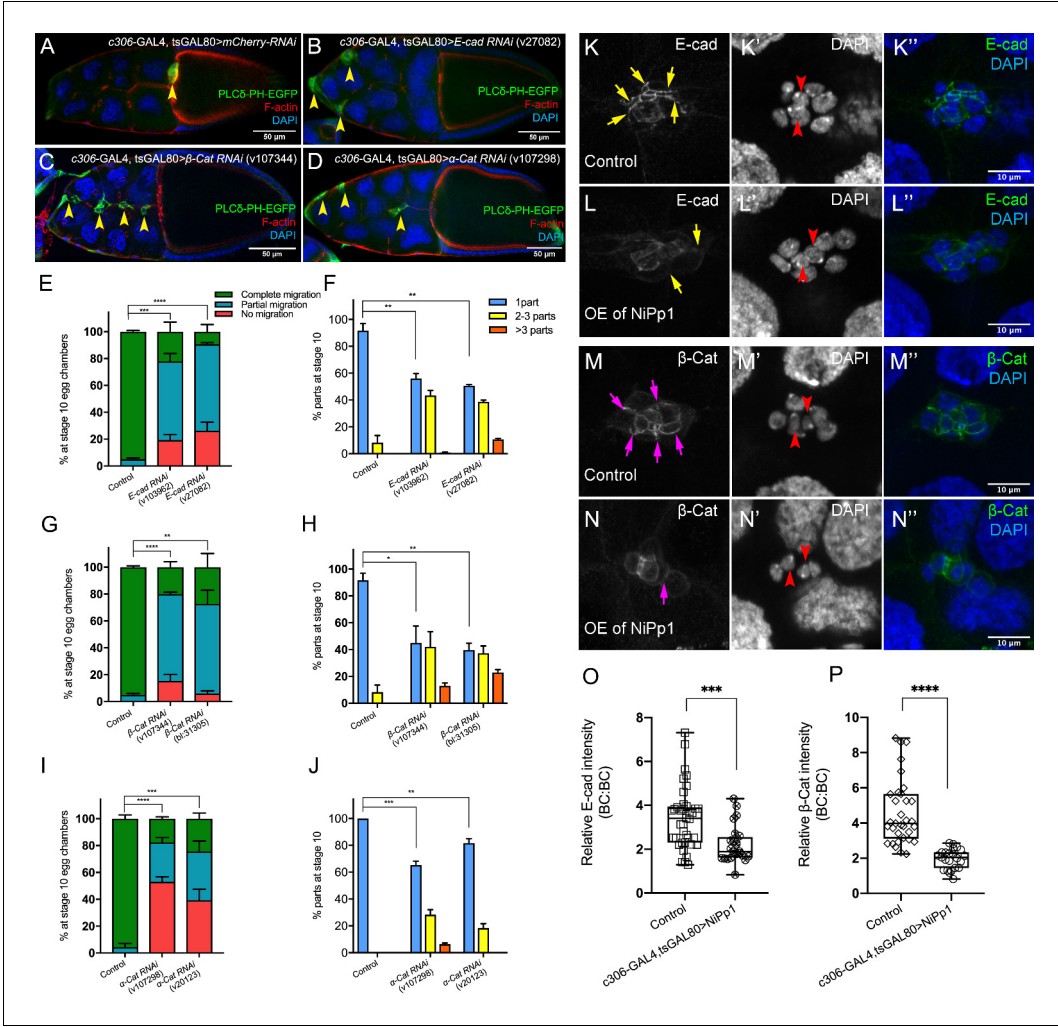

**Figure 4.** The cadherin-catenin complex is required for the collective cohesion of the migrating border cell cluster and is regulated by Pp1. (**A–J**) Knocking down *E-Cad*, *β-Cat* or *α-Cat* by RNAi disrupts border cell cluster migration and cohesion. Images of stage 10 egg chambers stained for phalloidin to label F-actin (red) and DAPI to label nuclei (blue). Border cells (arrowheads) express the membrane marker PLCδ-PH-EGFP (green). (**E–J**) Quantification of border cell migration (**E, G, I**) and cluster cohesion (**F, H, J**) in stage 10 control and *E-Cad-RNAi* (**E, F**), *β-Cat-RNAi* (**G, H**) and α*-Cat-RNAi* (**I, J**) egg chambers. The controls for *E-Cad* and *β-Cat-RNAi* are identical, but shown on separate graphs (**E–H**) for clarity; a separate matched control is shown for *α-Cat* RNAi (**I, J**). Error bars represent SEM in three experiments, each trial assayed n ≥ 27 egg chambers (total n ≥ 93 for *each genotype*). *p<0.05; **p<0.01; ***p<0.001; ****p<0.0001, unpaired two-tailed *t* test. (**E, G, I**) Quantification of border cell migration, shown as the percentage of egg chambers with complete (green), partial (blue), or no (red), border cell migration. (**F, H, J**) Quantification of cluster cohesion, shown as the percentage of border cells found as a single unit (one part) or split into multiple parts (2–3 parts or >3 parts) in control versus RNAi egg chambers. (**K–N''**) Representative images showing the E-Cad (white in K, L; green in K'', **L''**) and β-cat (white in M, N; green in M'', **N''**) protein expression pattern in control and NiPp1 overexpressing (OE) border cells. Border cells were co-stained for DAPI to mark nuclei (white in K', L', M', N'; blue in K'', **L''**, **M''**, **N''**). Images were generated from merged *z*-sections. The enriched levels of E-Cad (**K, L**) and β-cat (**M, N**) between border cells (border cell-border cell contacts) are marked by yellow and magenta arrows, respectively. The central polar cells are indicated by red arrowheads (**K', L', M', N'**). (**O, P**) Quantification of relative E-Cad (**O**) and β-Cat (**P**) protein intensity levels in control and NiPp1 overexpressing border cell clusters shown as box-and-whiskers plots (see *Figure 1* legend for details of plot). For E-Cad, 39 border cell-border cell contacts from eight matched control clusters and 24 border cell-border cell contacts from 16 NiPp1 clusters were measured. For β-Cat, 33 border cell-border cell contacts from seven matched control clusters and 23 border cell-border cell contacts from 15 NiPp1 clusters were measured. ***p<0.001, ****p<0.0001, unpaired two-tailed *t* test. All genotypes are listed in *Table 2*.
The online version of this article includes the following figure supplement(s) for figure 4:

*Figure 4 continued on next page*

**Figure supplement 1.** RNAi for cadherin-catenin reduces endogenous levels of the specifically targeted gene.

## NiPp1 inhibits the function of Pp1 catalytic subunits in border cells

NiPp1 is a specific inhibitor of Pp1c activity in vitro as well as in vivo (*Winkler et al., 2015*; *Parker et al., 2002*; *Van Eynde et al., 1995*). *Drosophila* has four Pp1c subunit genes (*Dombrádi et al., 1993*; *Dombrádi et al., 1990*), whereas humans have three genes (*Lin et al., 1999*). Pp1α−96A, Flapwing (Flw), and Pp1-87B transcripts are each expressed at moderate-to-high levels in the adult ovary, whereas Pp1-13C RNA is mainly detected in adult males (http://flybase.org/) (*Graveley et al., 2011*). We examined the localization of Pp1α−96A using a genomic fosmid transgene in which the open reading frame of Pp1α−96A is driven by its endogenous genomic regulatory regions and C-terminally tagged with GFP ('Pp1α−96A-GFP') (*Sarov et al., 2016*). Pp1α−96A-GFP was detected in the cytoplasm, with higher levels at the cortical membranes of border cells, follicle cells, the oocyte, and nurse cells (*Figure 2A–C*). Endogenous Flw, as visualized using a functional in-frame YFP protein trap (*Yamamoto et al., 2013*) ('Flw-YFP'), was also expressed ubiquitously during the stages in which border cells migrate (*Figure 2D–F*). Specifically, Flw-YFP was enriched at the cell cortex and cytoplasm of all cells, including border cells. Due to a lack of specific reagents, we were unable to determine whether Pp1-87B or Pp1-13C proteins are present in border cells. Therefore, at least two Pp1c subunit proteins are expressed in border cells throughout their migration.

We next determined whether NiPp1 specifically inhibited Pp1c activity in border cells. Overexpression of each of the four *Drosophila* Pp1c subunits individually did not impair border cell migration (*Figure 2—figure supplement 1A–D*). When co-expressed with NiPp1, two of the catalytic subunits, Pp1α−96A and Pp1-87B, strongly suppressed the migration defects caused by NiPp1, with 90% (NiPp1 + Pp1α−96A) and 75% (NiPp1 + Pp1-87B) of border cells now reaching the oocyte compared to 40% with NiPp1 alone (NiPp1 + RFP; *Figure 2G*; *Figure 2—figure supplement 1F–H*). Co-expression of Pp1α−96A and Pp1-87B partially suppressed the NiPp1-induced cluster fragmentation, leading to 55% (NiPp1 + Pp1α−96A) and 65% (NiPp1 + Pp1-87B) of border cell clusters now found intact compared to ~10% with NiPP1 alone (NiPp1 + RFP; *Figure 2H*; *Figure 2—figure supplement 1F–H*). Flw and Pp1-13C only mildly suppressed the NiPp1-induced cluster splitting and migration defects (*Figure 2G,H*; *Figure 2—figure supplement 1I,J*). The observed phenotypic suppressions were likely due to titration of NiPp1 inhibitory activity by excess Pp1c protein, in agreement with previous studies in *Drosophila* (*Bennett et al., 2003*; *Parker et al., 2002*). Partial suppression could be due to levels of overexpressed Pp1c or effectiveness of the respective Pp1c to titrate NiPp1 in border cells. Co-expression of a human Pp1c homolog ('hPPP1CC') fully suppressed the NiPp1-induced phenotypes and did not disrupt migration when expressed on its own (*Figure 2G,H*; *Figure 2—figure supplement 1E,K*). hPPP1CC has high homology to Pp1-87B (93% identical, 96% similar), Pp1α−96A (89% identical, 94% similar), and Pp1-13C (91% identical, 95% similar), with lower homology to Flw (84% identical, 91% similar), although further analysis through the DIOPT *Drosophila*RNAi Screening Center Integrative Ortholog Prediction Tool) database suggests higher homology to Pp1-87B and Pp1α−96A (http://flybase.org/) (*Hu et al., 2011*). The suppression by multiple Pp1 proteins and full suppression by hPPP1CC suggests that Pp1 catalytic subunit genes have overlapping functions in border cells.

To better understand how NiPp1 inhibits Pp1 activity in border cells, we next analyzed the subcellular localization of Flw-YFP and Pp1α−96A-GFP when NiPp1 was co-expressed. Expression of HA-tagged NiPp1 alone was itself predominantly nuclear, with low expression in the cytoplasm (*Figure 2—figure supplement 2A–A''*). Pp1α −96A-GFP and Flw-YFP normally localize to the cortical membrane and cytoplasm of border cells (*Figure 2A–F*). Upon co-expression with NiPp1, however, Flw-YFP and Pp1α −96A-GFP were now primarily localized to border cell nuclei along with NiPp1 (HA-tagged NiPp1; *Figure 2—figure supplement 2B–C''*). These results suggest that ectopic NiPp1, in addition to directly inhibiting Pp1c activity also sequesters PP1 catalytic subunits in the nucleus (*Winkler et al., 2015*; *Parker et al., 2002*; *Trinkle-Mulcahy et al., 2001*).

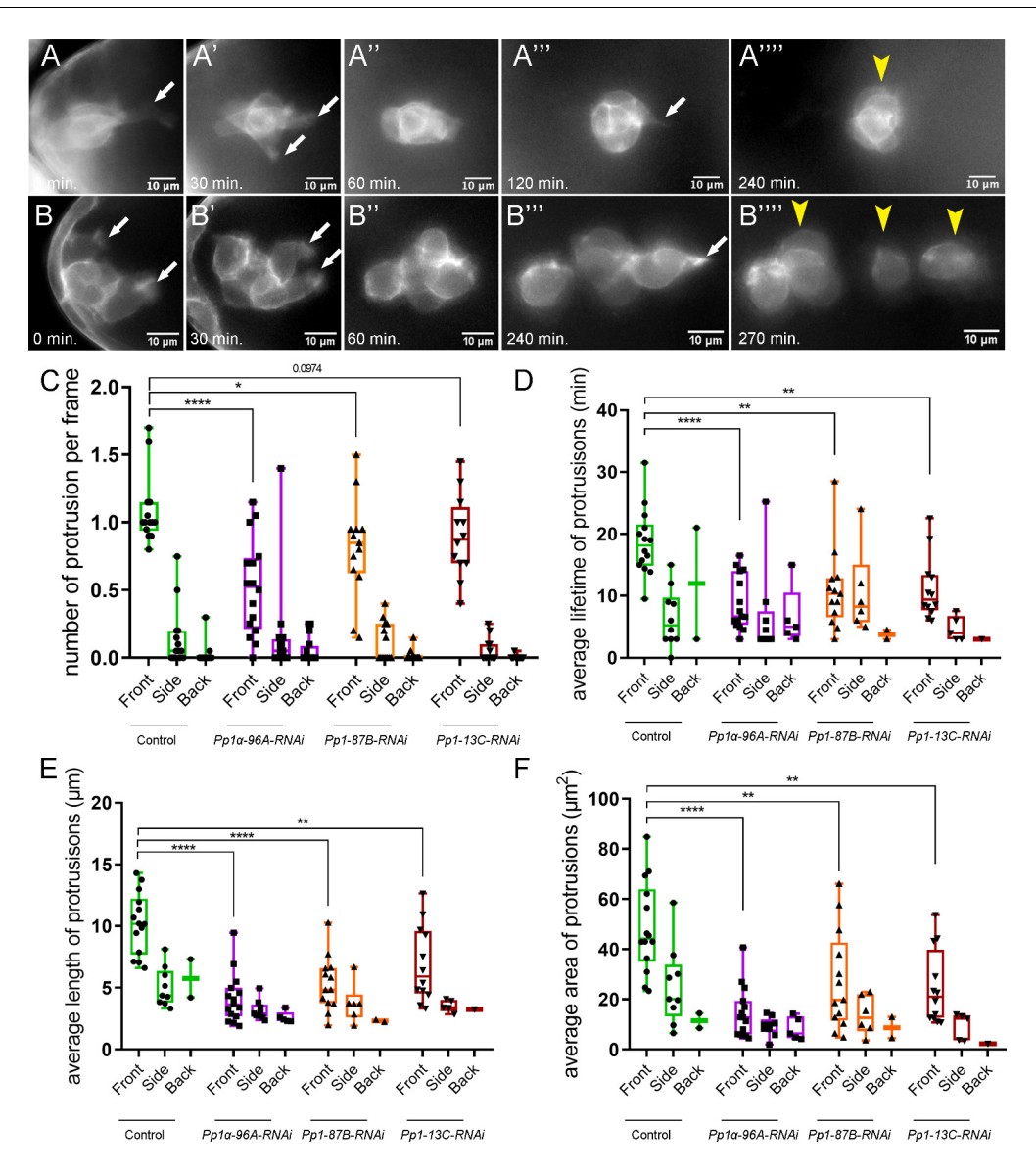

**Figure 5.** Pp1c is required for normal border cell protrusion dynamics. (A–B"") Frames from a matched control (*Video 6*; A–A"") and a *Pp1α-96A-RNAi* (*Video 8*; B–B"") showing the migrating border cell cluster expressing the membrane marker PLCδ-PH-EGFP. Time in min. Arrows indicate protrusions, arrowheads indicate cluster 'parts'. (C–F) Quantification of the number of protrusions per frame (C), average protrusion lifetime (D), average protrusion length (E), and average protrusion area (F) from videos of the indicated genotypes. Protrusions were defined as in *Figure 5—figure supplement 1A* and in the Materials and methods. For control, protrusions were measured in 14 videos (n = 51 front-directed protrusions, n = 15 side-directed protrusions, n = 2 back-directed protrusions); for *Pp1α-96A-RNAi*, protrusions were measured in n = 16 videos (n = 59 front protrusions, n = 19 side protrusions, n = 9 for back protrusions), for *Pp1-87B-RNAi*, protrusions were measured in 13 videos (n = 67 for front protrusions, n = 10 for side protrusions, n = 3 for back protrusions); for *Pp1-13C-RNAi*, protrusions were measured in 12 videos (n = 61 front protrusions, n = 9 side protrusions, n = 1 back protrusion). Data are presented as box-and-whiskers plots (see *Figure 1* legend for details of plot). *p<0.05, **p<0.01, ***p<0.001, ****p<0.0001, unpaired two-tailed *t* test. All genotypes are listed in *Table 2*.

The online version of this article includes the following figure supplement(s) for figure 5:

**Figure supplement 1.** Additional quantification of protrusion dynamics and Rac activity in Pp1-inhibited and α-Cat-RNAi border cells.

## Pp1c genes are required for border cell cluster migration and cohesion

To determine whether Pp1 catalytic activity itself is required for border cell migration, we next downregulated the *Pp1c* genes by driving the respective UAS-RNAi lines with *c306*-GAL4 (*Figure 3A–D*). RNAi lines that target 3 of the four catalytic subunits (*Pp1α−96A*, *Pp1-87B*, and *Pp1-13C*) strongly disrupted border cell migration (*Figure 3B–E*). The majority of *Pp1c* RNAi border cells either did not migrate ('no migration') or stopped along the migration pathway ('incomplete migration'; *Figure 3E*). *Pp1α−96A*-RNAi in particular, caused a significant fraction of border cells to fail to migrate at all, likely due to a failure to delaminate from the epithelium (~15%; *Figure 3E*). Knockdown of *Pp1c* genes also caused ≥50% of border cell clusters to dissociate into multiple sub-clusters and single cells (*Figure 3B–D,F*). Using live time-lapse imaging, we confirmed that decreased levels of Pp1α−96A, Pp1-87B, and Pp1-13C by RNAi altered border cell migration and caused cells to split from the main cluster (*Figure 3G*; *Videos 6*, *7*, *8*, *9* and *10*). Some *Pp1α−96A*-RNAi border cells did not delaminate from the epithelium during the course of imaging (*Figure 3—figure supplement 1A*; *Video 8*). Multiple *flw* RNAi lines (see Materials and methods) did not impair migration or cluster cohesion when expressed in border cell clusters. However, RNAi does not always fully knock down gene function in cells (*Mohr et al., 2014*). As complete loss of *flw* is homozygous lethal, we generated border cells that were mosaic mutant for the strong loss of function allele *flw^FP41* (*Sun et al., 2011*). Mosaic *flw^FP41* border cell clusters were typically composed of a mixture of wild-type and mutant cells that frequently fell apart, with ~90% splitting into two or more parts (*Figure 3H–I*; *Figure 3—figure supplement 1B–B"*). In egg chambers with *flw* mutant border cells, 40% of border cell sub-clusters did not delaminate or migrate at all ('no migration') whereas 20% partially migrated but did not reach the oocyte (*Figure 3H–H",J*; *Figure 3—figure supplement 1B–B"*). NiPp1 expression results in more severe phenotypes than RNAi knockdown, or loss, of individual *Pp1c* genes, at least with respect to migration and cluster cohesion, suggesting that Pp1c subunits have both distinct and overlapping functions. In particular, Pp1α−96A and Flw appear to function in border cell delamination, whereas all four subunits likely promote migration and cluster cohesion.

## Pp1 promotes cadherin-catenin complex levels and adhesion of border cells

One of the strongest effects of decreased Pp1c activity was the dissociation of border cells from the cluster. In many cell collectives, cadherins critically mediate the attachment of individual cells to each other during migration, although other cell-cell adhesion proteins can also contribute (*Friedl and Mayor, 2017*; *Collins and Nelson, 2015*). The cadherin-catenin complex members E-Cadherin (*Drosophila* Shotgun; Shg), β-Catenin (*Drosophila* Armadillo; Arm) and α-Catenin are all required for border cell migration (*Cai et al., 2014*; *Niewiadomska et al., 1999*; *Sarpal et al., 2012*; *Desai et al., 2013*). E-Cadherin, in particular, is required for traction of border cells upon the nurse cell substrate, for producing overall front-rear polarity within the cluster, and for attachment of border cells to the central polar cells (*Cai et al., 2014*; *Niewiadomska et al., 1999*). Complete loss of

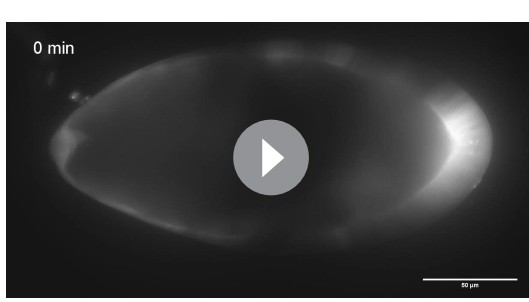

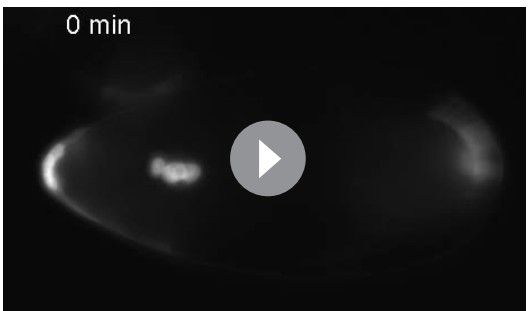

**Video 1.** Control (*c306*-GAL4/+; *UAS*-mCherry-Jupiter/+) egg chamber showing normal border cell migration. Frames were acquired every 3 min with a 20x objective. Anterior is to the left.
https://elifesciences.org/articles/52979#video1

**Video 2.** NiPp1 overexpressing (*c306*-GAL4/+; *UAS*-mCherry-Jupiter/+; *UAS*-NiPp1/+) egg chamber showing the migration defect and splitting phenotype. Frames were acquired every 3 min with a 20x objective. Anterior is to the left.
https://elifesciences.org/articles/52979#video2

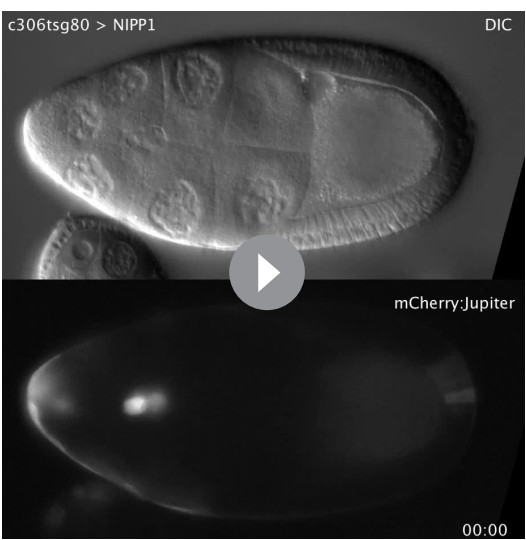

**Video 3.** Representative time-lapse video of a stage 9 NiPp1 overexpressing (*c306*-GAL4,tsGAL80/+; *UAS*-mCherry-Jupiter/+; *UAS*-NiPp1/+) egg chamber with DIC channel. Frames were acquired every 2 min with a 20x objective. Time is in hr:min. Anterior is to the left.
https://elifesciences.org/articles/52979#video3

**Video 4.** Representative time-lapse video of a stage 9 NiPp1 overexpressing (*c306*-GAL4,tsGAL80/+; *UAS*-mCherry-Jupiter /+;*UAS*-NiPp1/+) egg chamber with DIC channel. Frames were acquired every 2 min with a 20x objective. Time is in hr:min. Anterior is to the left.
https://elifesciences.org/articles/52979#video4

cadherin-catenin complex members in border cells prevents any movement between nurse cells (*Niewiadomska et al., 1999*; *Sarpal et al., 2012*; *Desai et al., 2013*). This has precluded a definitive analysis of whether all, or some, complex members promote adherence of border cells to the polar cells and/or to other border cells.

To determine whether adhesion of border cells to the cluster requires a functional cadherin-catenin complex, we used *c306*-GAL4 to drive RNAi for each gene in all cells of the cluster (*Figure 1—figure supplement 2B*). Multiple non-overlapping RNAi lines for *E-Cadherin*, *β-Catenin*, and *α-Catenin* each reduced the respective endogenous protein levels and disrupted border cell migration, in agreement with previous results that used mutant alleles (*Figure 4A–E,G,I*; *Figure 4—figure supplement 1A–H'*; *Video 11*; *Niewiadomska et al., 1999*; *Sarpal et al., 2012*; *Pacquelet and Rørth, 2005*). Importantly, RNAi knockdown for each of the cadherin-catenin complex genes, driven by *c306*-GAL4, resulted in significant fragmentation of the border cell cluster compared to controls. *E-Cadherin* (40–50%) and *β-Catenin* (55–80%) RNAi lines exhibiting stronger, while *α-Catenin* RNAi lines exhibited milder (~20–30%), cluster fragmentation (*Figure 4A–D,F,H,J*; *Video 11*). Dissociated RNAi border cells could localize to the side of the egg chamber (*Figure 4B,D*), although others remained on the normal central migration pathway (*Figure 4C,D*). While *α-Catenin* RNAi knockdown in polar cells alone (*upd*-GAL4) caused border cell cluster splitting and migration defects, this effect was significantly milder than

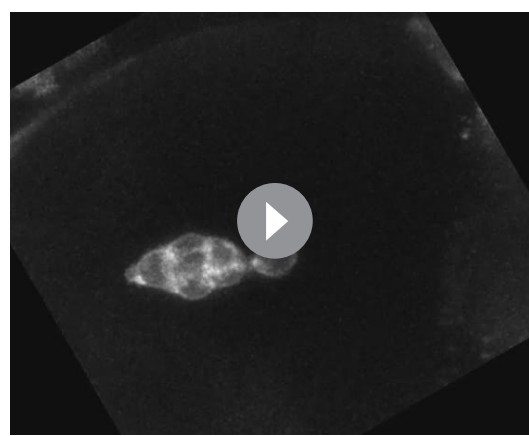

**Video 5.** NiPp1 overexpressing (*slbo*-GAL4/+; UAS-PLCδ-PH-EGFP/UAS-NiPp1) egg chamber showing the loss of a membrane attachment between one border cell and the rest of the border cell cluster. Anterior is to the left.
https://elifesciences.org/articles/52979#video5

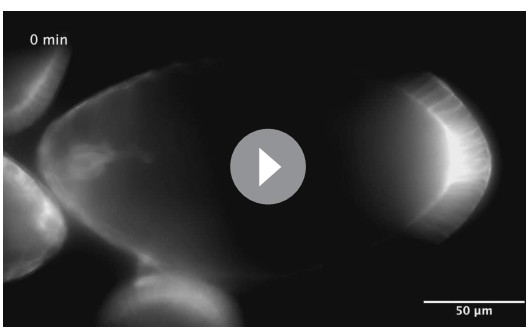

**Video 6.** Control (*c306*-GAL4,tsGAL80/+; UAS- PLCδ-PH-EGFP/+) egg chamber showing normal border cell migration. Frames were acquired every 3 min with a 20x objective. Anterior is to the left.
https://elifesciences.org/articles/52979#video6

**Video 7.** Representative time-lapse video of a stage 9 *Pp1α−96A* RNAi (*c306*-GAL4,tsGAL80/+; *v27673*/+; PLCδ-PH-EGFP /+) egg chamber. Frames were acquired every 3 min with a 20x objective. Anterior is to the left.
https://elifesciences.org/articles/52979#video7

the effects of *α-Catenin* knockdown in both polar cells and border cells using *c306*-GAL4 (compare *Figure 4I,J* to *Figure 4—figure supplement 1I, J*). These results indicate that the cadherin-catenin complex keeps border cells attached to each other and to the polar cells, which in turn maintains a cohesive cluster.

We next wanted to determine whether Pp1 regulated these adhesion proteins in border cells. We analyzed the levels and localization of E-Cadherin and β-Catenin at cell-cell contacts in NiPp1-expressing border cell clusters that were still intact or loosely connected (*Figure 4K–P*). In wild-type clusters, E-Cadherin and β-Catenin are highly enriched at cell contacts between border cells (BC-BC) and between border cells and polar cells (BC-PC; *Figure 4K–K",M–M"*). NiPp1-expressing border cell clusters exhibited reduced levels of E-Cadherin and β-Catenin at most BC-BC contacts (*Figure 4L–L",N–N"*). Pp1-inhibited polar cells generally retained E-Cadherin and β-Catenin, which was higher compared to border cells (*Figure 4L–L",N–N"*). We quantified the relative levels of E-Cadherin (*Figure 4O*) and β-Catenin (*Figure 4P*) at BC-BC contacts in control versus NiPp1 clusters, normalized to the levels of those proteins at nurse cell-nurse cell junctions. Both E-Cadherin and β-Catenin were reduced by almost half compared to matched controls. These data together suggest that Pp1 activity regulates cadherin-catenin proteins at cell-cell contacts, which contributes to adhesion of border cells within the cluster.

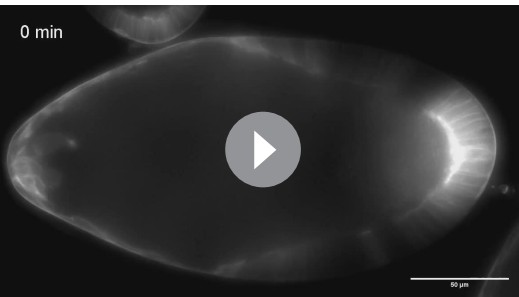

**Video 8.** Another representative time-lapse video of a stage 9 *Pp1α−96A* RNAi (*c306*-GAL4,tsGAL80/+; *v27673*/+;UAS-PLCδ-PH-EGFP/+) egg chamber. Frames were acquired every 3 min with a 20x objective. Anterior is to the left.
https://elifesciences.org/articles/52979#video8

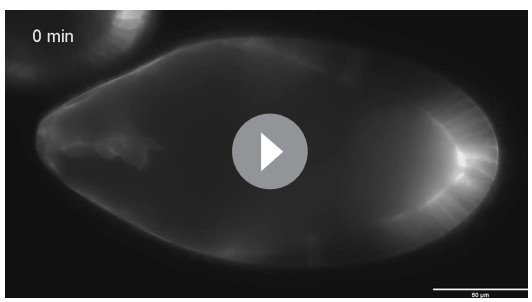

**Video 9.** Representative time-lapse video of a stage 9 *Pp1-13C* RNAi (*c306*-GAL4,tsGAL80/+;*v29058*/+;UAS-PLCδ-PH-EGFP/+) egg chamber. Frames were acquired every 3 min with a 20x objective. Anterior is to the left.
https://elifesciences.org/articles/52979#video9

## Pp1 activity promotes protrusion dynamics but is dispensable for directional migration

Border cells with impaired Pp1 activity migrated significantly slower than control clusters (*Figures 1M* and *3G*), suggesting that border cell motility was altered. Migrating cells form actin-rich protrusions at the front, or leading edge, which help anchor cells to the migratory substrate and provide traction for forward movement (*Ridley, 2011*; *Caswell and Zech, 2018*). In collectives, protrusive leader cells also help sense the environment to facilitate directional migration (*Mayor and Etienne-Manneville, 2016*). Border cells typically form one or two major protrusions at the cluster front (*Prasad and Montell, 2007*; *Poukkula et al., 2011*; *Wang et al., 2010*; *Figure 5A–A""*,C; *Figure 5—figure supplement 1A*; *Video 6*). Pp1-inhibited border cells (Pp1c RNAi) still extended forward-directed protrusions (*Figure 5A–C*; *Videos 7*, *8*, *9* and *10*). Additionally, the numbers, lifetimes, lengths and areas of side- and back-directed protrusions were not generally increased in Pp1-inhibited border cell clusters compared to control (*Figure 5C–F*; *Figure 5—figure supplement 1B,C*). However, the number of protrusions produced at the front of the cluster was reduced in Pp1 RNAi border cells (range of 0.5–0.85 mean protrusions per frame, all genotypes) compared to control (1.0 mean protrusions per frame; *Figure 5C*). Additionally, the lifetimes of Pp1 RNAi forward-directed protrusions were reduced (*Figure 5D*). Control protrusions at the cluster front had a lifetime of ~18 min, whereas Pp1-inhibited front protrusions persisted for 5–10 min. These short-lived Pp1 RNAi protrusions were also reduced in length, from a third to half the size of control front-directed protrusions (*Figure 5E*; *Figure 5—figure supplement 1B*). Further, Pp1-inhibited front protrusions were smaller, with a mean area of ~10–20 $\mu m^2$ compared to the control mean of ~40 $\mu m^2$ (*Figure 5F*; *Figure 5—figure supplement 1C*). Thus, Pp1 activity promotes normal protrusion dynamics, including the number, lifetime and size of front-directed protrusions.

The majority of NiPp1 and Pp1c RNAi border cells followed the normal migratory pathway down the center of the egg chamber between nurse cells, even when cells broke off from the main cluster (*Figure 1H,L–L"* and *3B–D*; *Videos 2*, *3*, *4*, *5* and *7*, *8*, *9*, *10*). Moreover, in Pp1 RNAi border cells, front-directed protrusions still formed though with altered dynamics. These observations together suggest that Pp1 activity is not required for directional chemotactic migration. To further test this idea, we made use of a Förster Resonance Energy Transfer (FRET) activity reporter for the small GTPase Rac. Normally, high Rac-FRET activity occurs at the cluster front during early migration in response to guidance signals from the oocyte, and correlates with protrusion extension (*Figure 5—figure supplement 1D*; *Wang et al., 2010*). Under conditions of PP1-inhibition, the most severely affected clusters fall apart, sometimes on different focal planes, making it difficult to interpret Rac-FRET signal. We therefore measured global Rac-FRET only in those NiPp1-expressing border cell clusters that remained intact. We detected elevated Rac-FRET activity in NiPp1 border cells similar to control, indicating that Rac activity was largely preserved although with slightly elevated levels (*Figure 5—figure supplement 1D,E*). In sum, these data indicate that Pp1 activity influences protru-

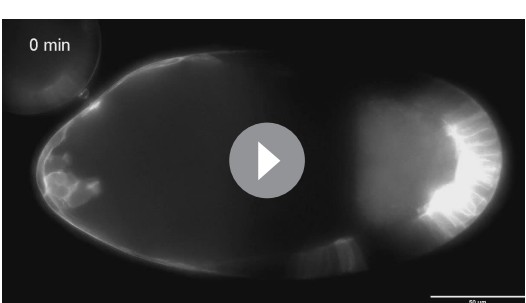

**Video 10.** Representative time-lapse video of a stage 9 *Pp1-87B* RNAi (*c306*-GAL4,tsGAL80/+; *v35024/+;*UAS-PLCδ-PH-EGFP/+) egg chamber. Frames were acquired every 3 min with a 20x objective. Anterior is to the left.
https://elifesciences.org/articles/52979#video10

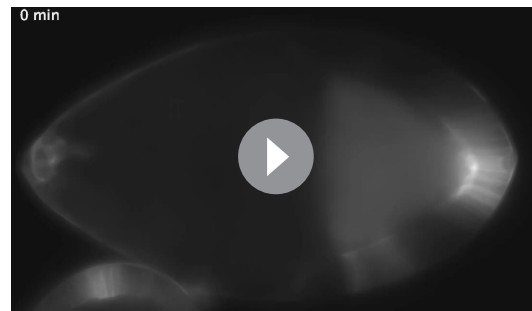

**Video 11.** Representative time-lapse video of a stage 9 *α-Cat* RNAi (*c306*-GAL4,tsGAL80/+; *v107298/+;* UAS-PLCδ-PH-EGFP/+) egg chamber. Frames were acquired every 3 min with a 20x objective. Anterior is to the left.
https://elifesciences.org/articles/52979#video11

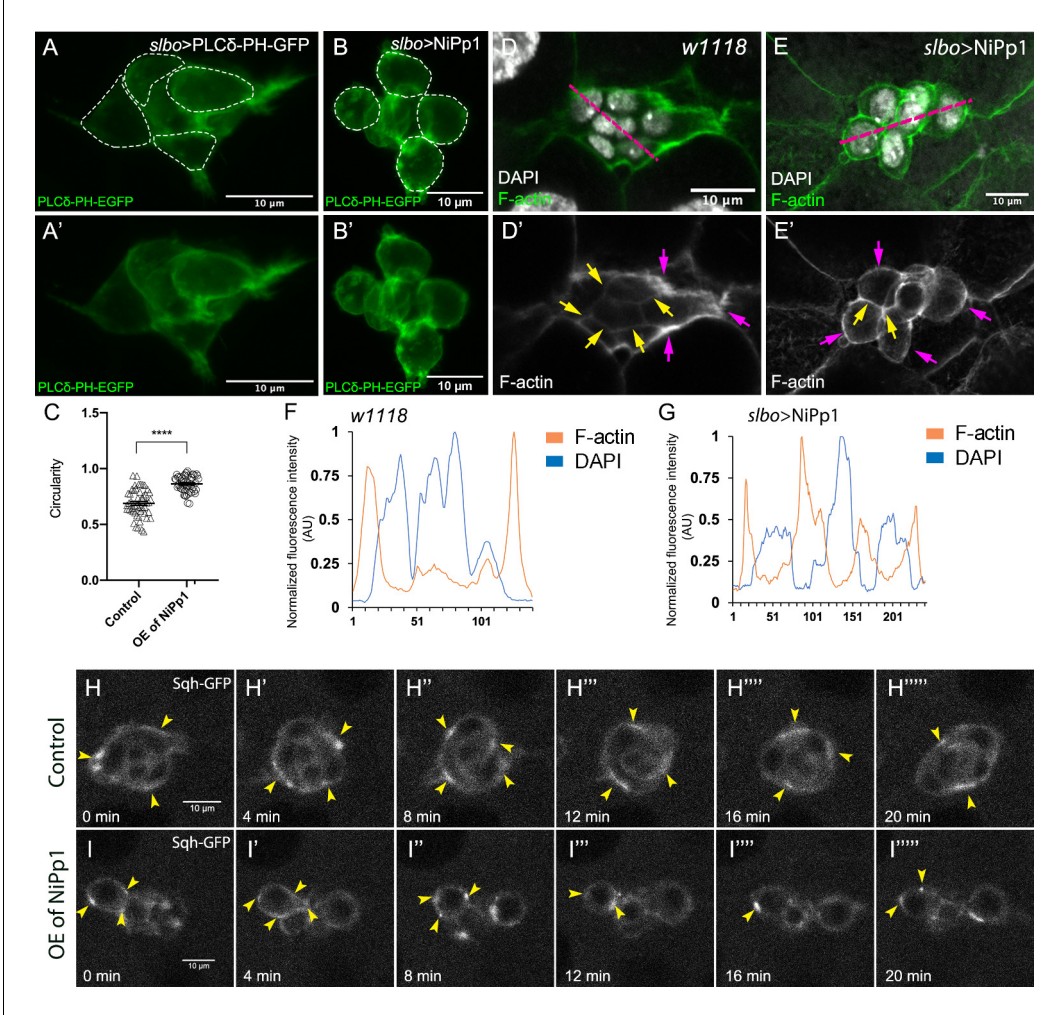

**Figure 6.** Pp1 activity promotes normal border cell shape and distribution of actomyosin in the border cell cluster. (A–C) Pp1 is required for border cell shape. (A–B') Examples of control (A, A') and NiPp1-expressing border cells (B, B'). Cell shape was visualized using the membrane marker PLCδ-PH-EGFP driven by *slbo*-GAL4 (green). Cells were outlined (A, B) and measured for circularity (C). (C) Control border cells are more elongated compared to NiPp1-expressing border cells (closer to 1.0, a perfect circle). Quantification of circularity, showing all data points and the mean; 51 control border cells and 57 NiPp1-expressing border cells were measured. ****p<0.0001, unpaired two-tailed *t* test. (D–G) Pp1 restricts high levels of F-actin to the border cell cluster periphery. Egg chambers were stained for phalloidin to detect F-actin (green in D, E; white in D', E') and DAPI to visualize nuclei (white in D, E). (D, D') Control wild-type border cells ($w^{1118}$) have higher F-actin at the cluster perimeter (magenta arrows) and low levels at cell-cell contacts inside the cluster (yellow arrows). (E, E') NiPp1 overexpression increases F-actin inside the cluster at cell contacts between border cells and at cell contacts between polar cells and border cells (yellow arrows). F-actin is relatively high on the outer surfaces of border cells (magenta arrows). (F, G) Plot profiles of normalized F-actin (orange) and DAPI (blue) fluorescence pixel intensity (AU, arbitrary units) measured along the lines shown in (D) and (E); similar results were obtained from additional border cell clusters (n = 11 for control and n = 8 for *slbo* >NiPp1). (H–I''''') Pp1 restricts Myo-II, as visualized by Sqh-GFP, to the cluster periphery in live border cells. Stills from confocal videos of Sqh-GFP in mid-staged border cells over the course of 20 min. Enriched Sqh-GFP is marked by arrowheads. Imaging gain and other acquisition parameters were the same, except that the range of z-stacks vary slightly. Similar patterns were observed for control in n = 8 movies and n = 10 for NiPp1 overexpression. (H–H''''') Control border cells (*Video 16*). (I–I''''') NiPp1 overexpression (*Video 17*) changes the dynamics of Sqh-GFP, with more Sqh-GFP located in individual border cells and at cell contacts between border cells. All genotypes are listed in *Table 2*.

The online version of this article includes the following figure supplement(s) for figure 6:

**Figure supplement 1.** Pp1 restricts the distribution of Myo-II to the cluster periphery during early migration.
**Figure supplement 2.** RNAi for cadherin-catenin alters the actomyosin pattern of the border cell cluster.

*Figure 6 continued on next page*

*Figure 6 continued*

**Figure supplement 3.** Myo-II is not required for cadherin-catenin enrichment at border cell-border cell contacts.

sion dynamics and cell motility, but does not appear to be critical for directional orientation of the cluster to the oocyte.

## Pp1 promotes border cell shape through collectively polarized F-actin and Myo-II

Migrating cells, including cell collectives, change shape to facilitate their movement through complex tissue environments (*Te Boekhorst et al., 2016*). Some cells maintain a single morphology, such as an elongated mesenchymal or rounded amoeboid shape, throughout migration, whereas other cells interconvert from one shape to another as they migrate. The border cell cluster overall is rounded, although individual border cells within the group appear slightly elongated (*Figure 6A,A'*; *Videos 1* and *6*; *Aranjuez et al., 2016*). However, NiPp1 border cells, whether present in small groups or as single cells, were visibly rounder than control border cells (*Figure 1H,L–L"*; *Videos 1–4*). We observed similar cell rounding when the *Pp1c* genes were knocked down by RNAi, although some border cells appeared more noticeably round than others (*Figures 3B–D* and *5B–B""*; *Videos 7–10*). To quantify these altered cell shapes, we expressed the membrane marker PLCδ-PH-EGFP to visualize individual cells within the cluster and measured 'circularity', which indicates how well a shape approaches that of a perfect circle (1.0; *Figure 6A–C*). Control border cells overall were slightly elongated with a mean of ~0.7, although the circularity of individual cells varied substantially (range of ~0.4 to 0.95), suggesting that border cells undergo dynamic shape changes during migration (*Figure 6C*). In contrast, NiPp1 border cells were rounder, with a mean of ~0.9, and exhibited less variation than control (range of ~0.7 to 1.0; *Figure 6C*).

The rounder cell shapes suggested that Pp1 inhibition alters the cortical cytoskeleton of the border cells. Wild-type border cells exhibit a marked enrichment of F-actin at the cluster periphery, whereas lower levels are detected inside the cluster at contacts between border cells (*Figure 6D,D', F*; *Video 12*; *Lucas et al., 2013*; *Wang et al., 2018*). Upon Pp1 inhibition, F-actin now accumulated around each individual border cell, especially at BC-BC membrane contacts, rather than just being enriched at outer cluster surfaces (*Figure 6E,E',G*; *Video 13*). Similarly, Myo-II as visualized by GFP-tagged Spaghetti Squash (Sqh-GFP), the *Drosophila* homolog of the myosin regulatory light chain (MRLC), is highly dynamic and normally concentrates in enriched foci at the outer periphery of live border cell clusters both during early (*Figure 6—figure supplement 1A–A""'*; *Video 14*) and later stages of migration (*Figure 6H–H""'*; *Video 16*;

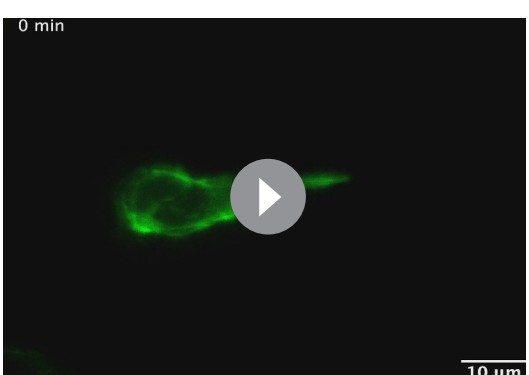

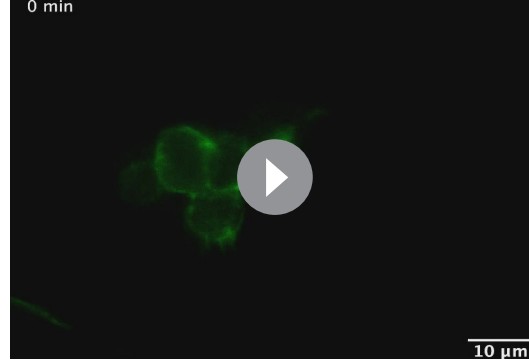

**Video 12.** Control (LifeAct-GFP/+) egg chamber showing the dynamics of F-actin with LifeAct-GFP, Frames were acquired every 2 min with a 40x water immersion objective. We observed similar dynamics in three videos. Anterior is to the left.
https://elifesciences.org/articles/52979#video12

**Video 13.** NiPp1 overexpressing (*slbo*-Gal4/+; UAS-NiPp1/LifeAct-GFP) egg chamber showing F-actin dynamics with LifeAct-GFP. Frames were acquired every 2 min with a 40x water immersion objective. We observed similar dynamics in three videos. Anterior is to the left.
https://elifesciences.org/articles/52979#video13

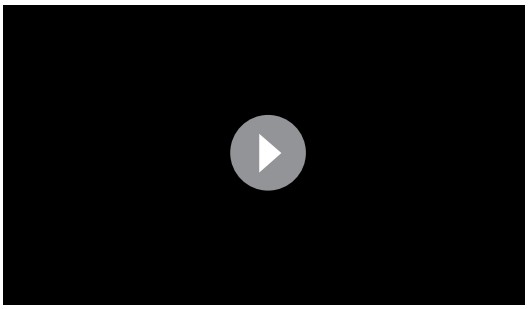

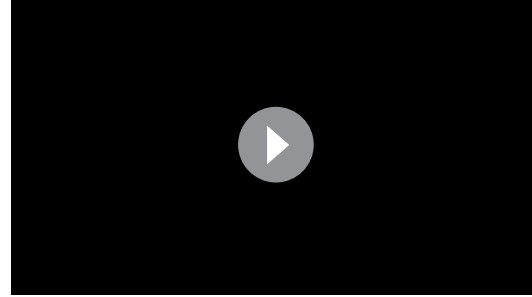

**Video 14.** Control (*c306*-GAL4, tsGAL80/+; Sqh-GFP/+) egg chamber showing normal Sqh-GFP dynamics in early migration. Frames were acquired every 1 min with a 40x water immersion objective, only a single focal plane is shown, with a 3D projection of the entire z-stack at the beginning and the end of the video. Similar patterns were observed in three videos. Anterior is to the left.

https://elifesciences.org/articles/52979#video14

**Video 15.** Representative NiPp1 overexpressing (*c306*-GAL4, tsGAL80/+; UAS-NiPp1/Sqh-GFP) egg Chamber showing the Sqh-GFP dynamics in early migration. Frames were acquired every 1 min with a 40x water immersion objective, only a single focal plane is shown, with a 3D projection of the entire z-stack at the beginning and the end of the video. Similar patterns were observed in four videos. Anterior is to the left.

https://elifesciences.org/articles/52979#video15

*Aranjuez et al., 2016*; *Combedazou et al., 2017*; *Majumder et al., 2012*). In NiPp1 border cells, however, Sqh-GFP was now present at cortical cell membranes in dynamic foci surrounding each border cell (or sub-cluster) rather than at the entire cluster periphery, both during early migration (*Figure 6—figure supplement 1B–B""'*; *Video 15*) and at mid-migration stages (*Figure 6I–I"'*; *Video 17*). Thus, inhibition of Pp1 converts collectively polarized F-actin and Myo-II to that characteristic of single migrating cells. As a result, individual border cells now have enriched and dynamic actomyosin localization consistent with elevated cortical contractility in single cells rather than at the collective level.

## Pp1 promotes actomyosin contractility in border cells through myosin phosphatase

Rok and other kinases phosphorylate the Myo-II regulatory light chain Sqh (*Vicente-Manzanares et al., 2009*). This leads to fully activated Myo-II, which then forms bipolar filaments, binds to F-actin, and promotes cell contractility. Given the altered distribution of Sqh-GFP when Pp1 was inhibited, we next analyzed the levels and

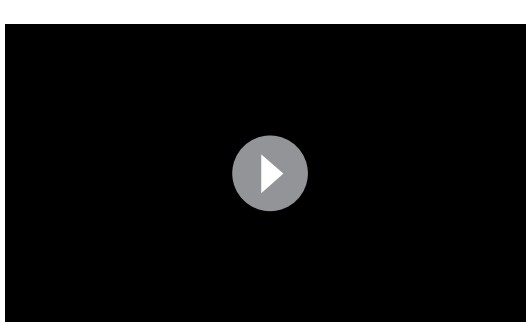

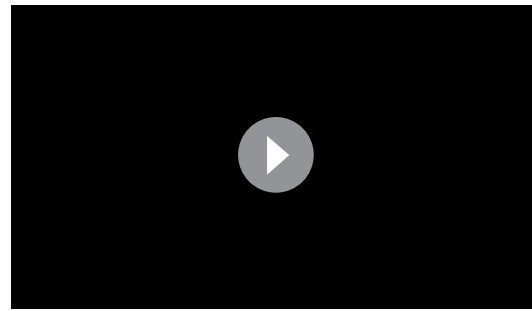

**Video 16.** Control (Sqh-GFP/+) egg chamber showing normal Sqh-GFP dynamics in mid-migration. Frames were acquired every 1 min with a 40x water immersion objective, only a single focal plane is shown, with a 3D projection of the entire z-stack at the beginning and the end of the video. Similar patterns were observed in five videos. Anterior is to the left.

https://elifesciences.org/articles/52979#video16

**Video 17.** Representative NiPp1 overexpressing (*c306*-GAL4, tsGAL80/+; UAS-NiPp1/Sqh-GFP) egg chamber showing the Sqh-GFP dynamics in mid-migration. Frames were acquired every 1 min with a 40x water immersion objective, only a single focal plane is shown, with a 3D projection of the entire z-stack at the beginning and the end of the video. Similar patterns were observed in six videos. Anterior is to the left.

https://elifesciences.org/articles/52979#video17

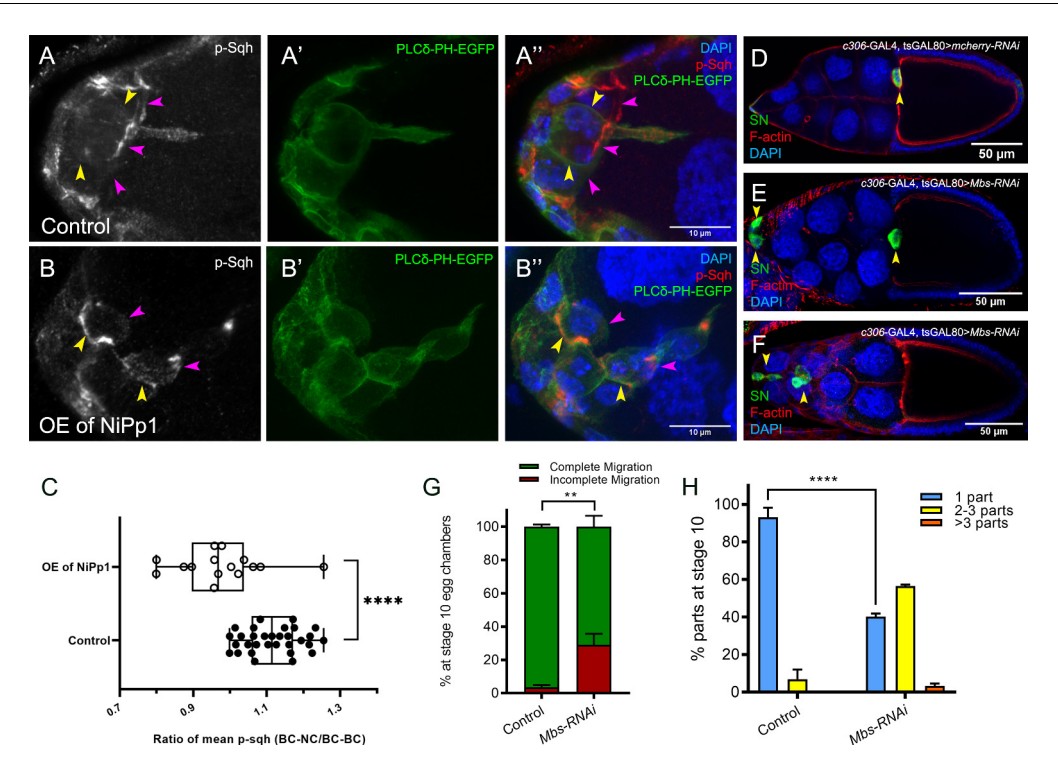

**Figure 7.** Pp1, through myosin phosphatase, promotes contractility of the cluster. (A–B'') Pp1 restricts Myo-II activation to the cluster periphery. Representative images showing p-Sqh localization (white in A, B; red in A'', (B'') and membrane GFP (PLCδ-PH-EGFP; green in A', (A'', B', B'') in control (A–A'') and NiPp1 overexpressing (B–B'') border cells; DAPI labels nuclei (blue in A'', (B''). There is an increase in p-Sqh levels (arrowheads) at the interface between border cells when NiPp1 is overexpressed. (C) Quantification of the mean pixel intensity of p-Sqh as a ratio of BC:NC/BC:BC. BC:NC stands for border cell-nurse cell interfaces, while BC:BC stands for border cell-border cell interfaces. N = 15 for control and n = 11 for NiPp1 overexpression. (D–H) Knocking down *Mbs* disrupts border cell migration and cluster cohesion. (D–F) Stage 10 control (D) and *Mbs* RNAi (E,F) egg chambers stained for SN to label border cells (green), phalloidin to label F-actin (red) and DAPI to label nuclei (blue). (G) Quantification of border cell cluster migration for matched control and *Mbs-RNAi*, shown as the percentage that did not complete (red), or completed (green) their migration to the oocyte (see *Figure 1I* for egg chamber schematic). (H) Quantification of cluster cohesion at stage 10, shown as the percentage of border cells found as a single unit (one part) or split into multiple parts (two parts, three parts,>3 parts) in control versus *Mbs-RNAi* border cells. (G, H) Each trial assayed n ≥ 61 egg chambers (total n ≥ 220 per genotype). **p<0.01; ****p<0.0001; unpaired two-tailed *t* test. All genotypes are listed in *Table 2*.

The online version of this article includes the following figure supplement(s) for figure 7:

**Figure supplement 1.** Expression of Mbs during border cell migration and specificity of *Mbs*-RNAi knockdown.
**Figure supplement 2.** Pp1 promotes moderate levels of RhoA activity in border cells.

distribution of active Myo-II. We used an antibody that recognizes phosphorylated Sqh at the conserved Ser-21 (mammalian MRLC Ser-19; 'p-Sqh') (*Majumder et al., 2012*). Control border cells exhibited p-Sqh signal primarily at the cluster periphery ('BC-NC' contacts; *Figure 7A–A''*). This pattern of p-Sqh closely resembles the pattern of Sqh-GFP in live wild-type border cells (*Figure 6H–H''''*; *Aranjuez et al., 2016*; *Majumder et al., 2012*; *Zeledon et al., 2019*; *Plutoni et al., 2019*). NiPp1 border cells, however, had high levels of p-Sqh distributed throughout the cluster including at internal BC-BC contacts (*Figure 7B–B''*), similar to Sqh-GFP in live NiPp1 border cells (*Figure 6I–I''''*). We measured the relative ratio of p-Sqh fluorescence intensity at BC-NC contacts versus BC-BC contacts in control and NiPp1 border cell clusters (*Figure 7C*). Control border cells had a higher p-Sqh ratio than NiPp1, indicating less p-Sqh signal at BC-BC contacts. These data support the idea that Pp1 inhibition elevates Myo-II activation within single border cells and at BC-BC contacts.

Myo-II undergoes cycles of activation and inactivation via phosphorylation and dephosphorylation, respectively, to generate dynamic cellular contraction in vivo (*Vicente-Manzanares et al., 2009*). We previously showed that waves of dynamic Myo-II maintain the collective morphology of border cells to facilitate movement through the egg chamber (*Aranjuez et al., 2016*). The myosin phosphatase complex consists of a Pp1c subunit and a specific regulatory subunit, the myosin binding subunit (Mbs; also called myosin phosphatase-targeting subunit [MYPT]), which together dephosphorylate Sqh and inactivate Myo-II (*Grassie et al., 2011*). Previously, we found that Mbs was required for border cell cluster delamination from the epithelium and cell shape (*Aranjuez et al., 2016*; *Majumder et al., 2012*), although cluster cohesion had not been explicitly assessed. We therefore wanted to determine whether myosin phosphatase contributed to the above-described Pp1 functions in cell shape, cluster cohesion and migration. First, we confirmed that Mbs transcript and protein were expressed in border cells throughout migration (*Figure 7—figure supplement 1A–F*). Mbs protein colocalized with Pp1c subunits near border cell membranes and in the cytoplasm (*Figure 7—figure supplement 1G–J*). In general, Mbs colocalized more extensively with Flw-YFP than with Pp1α−96A-GFP (*Figure 7—figure supplement 1G–J*).

Next, we analyzed the functions of Mbs in border cells. Border cells deficient for Mbs (*Mbs*-RNAi) were rounder than control border cells, exhibited incomplete migration (~30%), and dissociated from the cluster (60%) along the migration pathway (*Figure 7D–H*). The phenotypes observed with *Mbs*-RNAi were generally milder than those observed with Pp1-inhibition (either NiPp1 or *Pp1c*-RNAi; compare to *Figures 1I,J* and *3E,F*). This could be due to incomplete knockdown by *Mbs*-RNAi, although we observed significant decreases in the levels of endogenous Mbs (*Figure 7—figure supplement 1K–L"*). Alternatively, myosin phosphatase, through a complex of Mbs/Pp1c, could be one of multiple Pp1 complexes required for border cell cluster migration and cohesion (*see* Discussion). Nonetheless, these findings indicate that myosin phosphatase, a specific Pp1 complex, helps promote the normal cell morphology and collective cohesion of border cells, in addition to facilitating the successful migration of the border cells.

RhoA activates Rho-associated kinase (Rok), thus leading to activation of Myo-II (*Vicente-Manzanares et al., 2009*). We and others previously found that expression of constitutively-activated RhoA (*Drosophila* Rho1) causes markedly rounder border cells and alters the distribution of F-actin and Myo-II at cell-cell contacts between border cells (*Aranjuez et al., 2016*; *Combedazou et al., 2017*). We therefore investigated whether Pp1 regulated RhoA activity in migrating border cells. We used a FRET construct that was recently shown to specifically report RhoA activity in ovarian follicle cells (*Qin et al., 2017*). Inhibition of Pp1 by NiPp1 moderately increased the overall levels of Rho-FRET in intact border cell clusters compared to control border cells (*Figure 7—figure supplement 2A–C*). These data suggest a general upregulation of the RhoA pathway upon Pp1 inhibition.

## Interplay between cadherin-catenin adhesion and actomyosin dynamics

During cellular morphogenesis, the cadherin-catenin complex and actomyosin contractility can interact to influence cell-cell junction stability (*Mège and Ishiyama, 2017*; *Yap et al., 2018*; *Priya et al., 2015*; *Ratheesh et al., 2012*; *le Duc et al., 2010*). Given the effects of Pp1 inhibition on the cadherin-catenin complex, F-actin, and Myo-II, we asked whether the observed Pp1-dependent phenotypes were secondarily due to decreased adhesion and/or altered actomyosin contractility. Knockdown of *E-cadherin* or *β-catenin* by RNAi decreased the enrichment of F-actin and p-Sqh at the cluster periphery compared to controls (*Figure 6—figure supplement 2A–C', E–G,I–K*). This is in agreement with a recent study that observed decreased cortical Myo-II in live *E-cadherin*-RNAi border cells (*Mishra et al., 2019*). Despite this decrease in F-actin, migrating live *α-Catenin* RNAi border cells, while slower than control, extended protrusions with normal dynamics (*Figure 5—figure supplement 1F–J*). Interestingly, F-actin was also no longer enriched at the cluster periphery of *Sqh*-RNAi border cells (*Figure 6—figure supplement 2D,D',H*). Thus, F-actin enrichment at the cluster periphery requires both cadherin-catenin and Myo-II. Moreover, the cadherin-catenin complex promotes enriched activated Myo-II at the outer cluster.

Next, we asked if Myo-II was required for cadherin-catenin enrichment at border cell-border cell junctions. *Sqh*-RNAi border cells had normal levels of E-cadherin (*Figure 6—figure supplement 3A–B", E*) and normal to slightly higher levels of β-catenin (*Figure 6—figure supplement 3C–D", F*). Knockdown of *Sqh* did not disrupt distribution of E-cadherin or β-catenin at border cell-border cell

contacts (*Figure 6—figure supplement 3A–D''*). These data suggest that Myo-II is not a major regulator of the cadherin-catenin complex in border cells. The phenotypes observed with RNAi-mediated knockdown of the cadherin-catenin complex and Sqh are in contrast to those observed with Pp1 inhibition (e.g. *Figures 5–7*). These results are consistent with a more direct role for Pp1 activity in controlling collective versus single cell dynamics of actomyosin and cadherin-catenin in border cells.

## Discussion

To migrate collectively, cells need to coordinate and cooperate at the multicellular level. Individual cells within a group must remain together, maintain optimal cell shapes, organize motility of neighboring cells, and polarize. The mechanisms that globally orchestrate single cell behaviors within migrating cell collectives are still unclear. Here we report that Pp1 activity is a critical regulator of key intra- and intercellular mechanisms that together produce collective border cell migration. Loss of Pp1 activity, through overexpression of NiPp1 or Pp1c RNAi, switches border cells from migrating as a cohesive cluster to moving as single cells or in small groups (*Figure 8A*). A critical aspect of this switch is the redistribution of enriched F-actin and Myo-II to cell contacts between individual border cells, rather than at the cluster periphery, and a concomitant loss of adhesion between cells. We identified one key Pp1 phosphatase complex, myosin phosphatase, that controls collective-level myosin contraction (*Figure 8B*). Additional phosphatase complexes, through as-yet-unknown regulatory subunits, likely function in border cells to generate collective F-actin organization, maintain cell-cell adhesions, and potentially to restrain overall RhoA activity levels. Our results support a model in which balanced Pp1 activity promotes collective border cell cluster migration, and timely delamination from the epithelium, by coordinating single border cell motility and keeping the cells together (*Figure 8A*).

Many collectively migrating cells require a supracellular enrichment of actomyosin at the group perimeter to help organize their movement (*Shellard and Mayor, 2019*; *Shellard et al., 2018*; *Hidalgo-Carcedo et al., 2011*; *Reffay et al., 2014*). Active Myo-II is required for border cell collective detachment from the epithelium, cluster shape, rotational movement of the cluster, and normal protrusion dynamics (*Aranjuez et al., 2016*; *Combedazou et al., 2017*; *Majumder et al., 2012*; *Mishra et al., 2019*; *Fulga and Rørth, 2002*). We show here that Pp1 organizes collective-level Myo-II-contractility during border cell migration. Inhibition of Pp1 shifts the balance of dynamic activated Myo-II from the cluster-level to individual border cells, resulting in rounded, hyper-contractile border cells that dissociate from the cluster. The myosin-specific Pp1 complex, myosin phosphatase, directly dephosphorylates Sqh and inhibits Myo-II activation (*Grassie et al., 2011*). Depletion of Mbs, the myosin-binding regulatory subunit of myosin phosphatase, causes rounder border cells and fragmentation of the cluster. We previously found that Mbs-deficient border cells have significantly higher levels of phosphorylated Sqh (p-Myo-II) (*Majumder et al., 2012*). Thus, myosin phosphatase inhibits Myo-II activation to promote coordinated collective contractility of border cells. Myosin phosphatase is a downstream target of Rok, which phosphorylates and inhibits the Mbs subunit (*Kimura et al., 1996*). Consistent with loss of myosin phosphatase activity, Pp1-inhibition increases phosphorylated active Sqh in individual border cells within the cluster. Thus, myosin phosphatase, downstream of Rok, promotes elevated active Myo-II (p-Sqh/p-Myo-II) and cortical contraction of the entire collective (*Figure 8B*). Interestingly, expression of constitutively activated RhoA also induces cellular hypercontractility, resulting in amoeboid-like round border cells (*Aranjuez et al., 2016*; *Combedazou et al., 2017*; *Mishra et al., 2019*). RhoA activates Rok, which directly phosphorylates and activates the Myo-II regulatory subunit Sqh (*Amano et al., 1996*; *Matsui et al., 1996*). We observe somewhat elevated RhoA activity in the absence of Pp1 activity. Thus, Pp1 may also restrain the overall levels of RhoA activity in border cells through an unknown Pp1 complex, which would further promote the collective actomyosin contraction of border cells (*Figure 8B*).

Myo-II is activated preferentially at the cluster periphery and not between internal border cell contacts. Mbs and at least one catalytic subunit, Flw, localize uniformly in border cells, both on the cluster perimeter and between cells. Such uniform phosphatase distribution would be expected to dephosphorylate and inactivate Myo-II everywhere, yet phosphorylated Sqh is only absent from internal cluster border cell contacts. Rok phosphorylates and inactivates Mbs in addition to directly activating Myo-II (*Kimura et al., 1996*). Our previous results indicate that Rok localizes to the cluster perimeter similar to p-Sqh, but there appeared to be overall less Rok between border cells

**Table 2.** Genotypes for figures.

List of genotypes shown in the figures.

| Figure | Panel | Genotype |
|---|---|---|
| *Figure 1* | A-F | *w1118* |
| | G | *c306*-GAL4,tsGAL80/+ |
| | H | *c306*-GAL4,tsGAL80/+;*UAS*-NiPp1/+ |
| | K | *c306*-GAL4/+; *UAS*-Cherry:Jupiter /+ |
| | L | *c306*-GAL4/+; *UAS*-Cherry:Jupiter /+;*UAS*-NiPp1/+ |
| *Figure 1—figure supplement 1* | A | *c306*-GAL4,tsGAL80/+;*UAS*-GFP.nls/+ |
| | B | *slbo*-GAL4/+;*UAS*-GFP.nls/+ |
| *Figure 1—figure supplement 2* | B | *c306*-GAL4/+;*UAS*-PLCdelta-PH-EGFP/+ |
| | D-G | *slbo*-GAL4,*UAS*-mCD8-GFP/+; |
| | | *slbo*-GAL4,*UAS*-mCD8-GFP/+;*UAS*-NiPp1/+ |
| | H-K | *upd*-GAL4/+;*UAS*-mCD8.ChRFP/+ |
| | | *upd*-GAL4/+;*UAS*-NiPp1/+ |
| | L-N | *c306*-GAL4,tsGAL80/+ |
| | | *c306*-GAL4,tsGAL80/+;*UAS*-NiPp1/+ |
| *Figure 1—figure supplement 3* | A | *c306*-GAL4/+ (WT) |
| | B | *c306*-GAL4,tsGAL80/+;*UAS*-NiPp1/+ |
| *Figure 2* | A-C | FlyFos021765(pRedFlp-Hgr) (Pp1alpha-96A15346::2XTY1-SGFP-V5-preTEV-BLRP-3XFLAG)dFRT |
| | D-F | w[1118] PBac{681 .P.FSVS-1}flw [CPTI002264] |
| | G-H | *c306*-GAL4,tsGAL80/+;*UAS*-NiPp1/*UAS*-mCD8.ChRFP |
| | | *c306*-GAL4,tsGAL80/+;*UAS*-NiPp1/*UAS*-Pp1α−96A.HA |
| | | *c306*-GAL4,tsGAL80/+;*UAS*-NiPp1/*UAS*-Pp1-87B.HA |
| | | *c306*-GAL4,tsGAL80/+;*UAS*-NiPp1/*UAS*-Pp1-13C.HA |
| | | *c306*-GAL4,tsGAL80/+;*UAS*-NiPp1/*UAS*-Flw.3xHA |
| | | *c306*-GAL4,tsGAL80/+;*UAS*-hPPP1CC/+;*UAS*-NiPp1/ |
| *Figure 2—figure supplement 1* | A | *c306*-GAL4/+;*UAS*-Pp1α−96A.HA/+ |
| | B | *c306*-GAL4/+;*UAS*-Pp1-87B.HA/+ |
| | C | *c306*-GAL4/+;*UAS*-Pp1-13C.HA/+ |
| | D | *c306*-GAL4/+;*UAS*-Flw.3xHA/+ |
| | E | *c306*-GAL4/+;*UAS*-hPPP1CC/+ |
| | F-K | Same as *Figure 2*. G-H |

*Table 2 continued on next page*

*Table 2 continued*

| Figure | Panel | Genotype |
|---|---|---|
| *Figure 2—figure supplement 2* | A | *c306*-GAL4,tsGAL80/+;*UAS*-NiPp1/+ |
| | B | *slbo*-GAL4/+;*UAS*-NiPp1/Pp1alpha-96A-GFP |
| | C | w1118/Flw-YFP;*slbo*-GAL4/+;*UAS*-NiPp1/+ |
| *Figure 3* | A-D | *c306*-GAL4/+;*UAS*-mCherry RNAi/+ |
| | | *c306*-GAL4/+;*UAS*-Pp1α−96A RNAi/+ |
| | | *c306*-GAL4/+;*UAS*-Pp1-87B RNAi /+ |
| | | *c306*-GAL4/+;*UAS*-Pp1-13C RNAi/+ |
| | G | *c306*-GAL4,tsGAL80/+;*UAS*-mCherry RNAi/*UAS*-PLCdelta-PH-EGFP |
| | | *c306*-GAL4,tsGAL80/+;*UAS*-Pp1α−96A RNAi/+;*UAS*-PLCdelta-PH-EGFP/+ |
| | | *c306*-GAL4,tsGAL80/+;*UAS*-Pp1-87B RNAi /+;*UAS*-PLCdelta-PH-EGFP/+ |
| | | *c306*-GAL4,tsGAL80/+;*UAS*-Pp1-13C RNAi/+;*UAS*-PLCdelta-PH-EGFP/+ |
| | H-H'' | *P{w[+mC]=Ubi mRFP.nls}1, w[*], P{ry[+t7.2]=hsFLP}12 P{ry[+t7.2]=neoFRT}19A/flwFP41 FRT 19A* |
| *Figure 3—figure supplement 1* | A | Same as *Figure 3*. G |
| | B | *P{w[+mC]=Ubi mRFP.nls}1, w[*], P{ry[+t7.2]=hsFLP}12 P{ry[+t7.2]=neoFRT}19A/flwFP41 FRT 19A* |
| *Figure 4* | A-J | *c306*-GAL4,tsGAL80/+;*UAS*-mCherry RNAi/+ |
| | | *c306*-GAL4,tsGAL80/+;*UAS*-E-cad RNAi (VDRC:103962)/+ |
| | | *c306*-GAL4,tsGAL80/+;*UAS*-E-cad RNAi (VDRC:27082)/+ |
| | | *c306*-GAL4,tsGAL80/+;*UAS*-β-Cat RNAi (VDRC:107344)/+ |
| | | *c306*-GAL4,tsGAL80/+;*UAS*-β-Cat RNAi (VDRC:31305)/+ |
| | | *c306*-GAL4,tsGAL80/+;*UAS*-α-Cat RNAi (VDRC:107298)/+ |
| | | *c306*-GAL4,tsGAL80/+;*UAS*-α-Cat RNAi (VDRC:20123)/+ |
| | K-P | *w1118(control)* |
| | | *c306*-GAL4,tsGAL80/+;*UAS*-NiPp1/+ |

*Table 2 continued on next page*

*Table 2 continued*

| Figure | Panel | Genotype |
|---|---|---|
| *Figure 4—figure supplement 1* | A,C,E,G | *c306-GAL4/+;UAS-mCherry RNAi/+* |
| | B | *c306-GAL4/+;UAS-E-cad RNAi (VDRC:103962)/+* |
| | D | *c306-GAL4/+;UAS-β-Cat RNAi (VDRC:107344)/+* |
| | F | *c306-GAL4/+;UAS-α-Cat RNAi (VDRC:107298)/+* |
| | G | *c306-GAL4/+;UAS-β-Cat RNAi (BDSC:31305)/+* |
| | I-J | *c306-GAL4/+;UAS-mCherry RNAi/+* |
| | | *upd-GAL4,tsGAL80/+;UAS-α-Cat RNAi (VDRC:107298)/+* |
| | | *upd-GAL4/+;UAS-α-Cat RNAi (VDRC:20123)/+* |
| *Figure 5* | A | *c306-GAL4,tsGAL80/+;UAS-mCherry RNAi/UAS-PLCdelta-PH-EGFP* |
| | B | *c306-GAL4,tsGAL80/+;UAS-Pp1α−96A RNAi/+;UAS-PLCdelta-PH-EGFP/+* |
| | C-F | Same as *Figure 3*. G |
| *Figure 5—figure supplement 1* | B-C | Same as *Figure 3*. G |
| | D-E | yw; *slbo*-GAL4/*UAS*-Rac FRET (WT) and *slbo*-GAL4/*UAS*-Rac FRET; +/*UAS*-NiPp1 |
| | F-J | *c306-GAL4,tsGAL80/+;UAS-mCherry RNAi/UAS-PLCdelta-PH-EGFP* |
| | | *c306-GAL4,tsGAL80/+;UAS-α-Cat RNAi (VDRC:107298);UAS-PLCdelta-PH-EGFP/+* |
| *Figure 6* | A | *slbo*-GAL4/+;*UAS*-PLCdelta-PH-EGFP/+ |
| | B | *slbo*-GAL4/+;*UAS*-NiPp1/*UAS*-PLCdelta-PH-EGFP |
| | D,F | *w1118 (control)* |
| | E,G | *slbo*-GAL4/+;*UAS*-NiPp1/+ |
| | H | *c306-GAL4,tsGAL80/+;+/sqh-GFP (VDRC:318484)* |
| | I | *c306-GAL4,tsGAL80/+;UAS-NiPp1/sqh-GFP (VDRC:318484)* |
| *Figure 6—figure supplement 1* | A | Same as *Figure 6*. H |
| | B | Same as *Figure 6*. I |
| *Figure 6—figure supplement 2* | A,A',E,I | *c306-GAL4,tsGAL80/+;UAS-mCherry RNAi/+* |
| | B,B',F,J | *c306-GAL4,tsGAL80/+;UAS-E-cad RNAi (VDRC:103962)/+* |
| | C,C',G,K | *c306-GAL4/+;UAS-β-Cat RNAi (BDSC:31305)/+* |
| | D,D',H | *c306-GAL4,tsGAL80/+;UAS-sqh RNAi (VDRC:7916)/+* |

*Table 2 continued on next page*

*Table 2 continued*

| Figure | Panel | Genotype |
| --- | --- | --- |
| *Figure 6—figure supplement 3* | A,C | *c306*-GAL4,tsGAL80/+;*UAS*-mCherry RNAi/+ |
| | B,D | *c306*-GAL4,tsGAL80/+;*UAS-sqh* RNAi (VDRC:7916)/+ |
| *Figure 7* | A-A' | *c306*-GAL4,tsGAL80/+;*UAS*-PLCdelta-PH-EGFP/+ |
| | B-B' | *c306*-GAL4,tsGAL80/+;*UAS*-PLCdelta-PH-EGFP/UAS-NiPp1 |
| | D-H | *c306*-GAL4,tsGAL80/+;*UAS*-mCherry RNAi/+ |
| | | *c306*-GAL4,tsGAL80/+;*UAS*-Mbs RNAi/+ |
| *Figure 7—figure supplement 1* | D-F | *w1118* |
| | G-G'' | FlyFos021765(pRedFlp-Hgr) (Pp1alpha-96A15346::2XTY1-SGFP-V5-preTEV-BLRP-3XFLAG) dFRT |
| | I-I'' | w[1118] PBac{681 .P.FSVS-1}flw [CPTI002264] |
| | K | *c306*-GAL4,tsGAL80/+;*UAS*-mCherry RNAi/+ |
| | L | *c306*-GAL4,tsGAL80/+;*UAS*-Mbs RNAi/+ |
| *Figure 7—figure supplement 2* | A-A' | *slbo*-GAL4/UAS-Rho FRET; +/UAS-Rho FRET |
| | B-B' | *slbo*-GAL4/UAS-Rho FRET;*UAS*-NiPp1/*UAS*-Rho FRET |

(*Aranjuez et al., 2016*). Thus, spatially localized Rok could inhibit myosin phosphatase and activate Myo-II preferentially at the outer edges of the cluster (*Figure 8A*). Other mechanisms likely contribute to collective polarization of Myo-II. For example, during border cell detachment from the epithelium the polarity kinase Par-1 phosphorylates and inactivates Mbs at the cluster rear resulting in increased active Myo-II, whereas the Hippo pathway prevents accumulation of phosphorylated Myo-II between border cells (*Lucas et al., 2013*; *Majumder et al., 2012*).

Our data also support a role for Pp1 in controlling F-actin stability, dynamics, and spatial organization. Similar to the pattern of activated Myo-II, cortical F-actin is normally high at the cluster periphery, although low levels are found between border cells (*Ramel et al., 2013*; *Lucas et al., 2013*; *Wang et al., 2018*). Reduced Pp1 activity causes high levels of F-actin to redistribute from the cluster perimeter to surround entire cell cortices of individual border cells. In migrating cells, networks of F-actin produce forces essential for protrusion extension and retraction dynamics that generate forward movement (*Ridley, 2011*; *Caswell and Zech, 2018*). Further supporting a role for Pp1 in regulating F-actin, Pp1-inhibited border cells extend fewer protrusions with shorter lifetimes, resulting in altered motility patterns. How Pp1 promotes F-actin organization and dynamics is unknown. One possibility comes from the known function for Rok in regulating F-actin through the downstream effector LIM Kinase (LIMK) (*Julian and Olson, 2014*). LIMK phosphorylates and inhibits cofilin, an actin severing and depolymerizing factor (*Bravo-Cordero et al., 2013*). In border cells, cofilin restrains F-actin levels throughout the cluster and increases actin dynamics, resulting in normal cluster morphology and major protrusion formation (*Zhang et al., 2011*). Although cofilin dephosphorylation, and thus activation, is typically mediated by the dual-specificity phosphatase Slingshot (*Bravo-Cordero et al., 2013*), Pp1-containing complexes have been shown to dephosphorylate cofilin in a variety of cell types (*Huet et al., 2013*; *Ambach et al., 2000*; *Zhang et al., 2012*; *Oleinik et al., 2010*). Additionally, RhoA activates formin proteins such as Diaphanous, which nucleate actin to form long filaments (*Kühn and Geyer, 2014*). There are at least seven formin-related

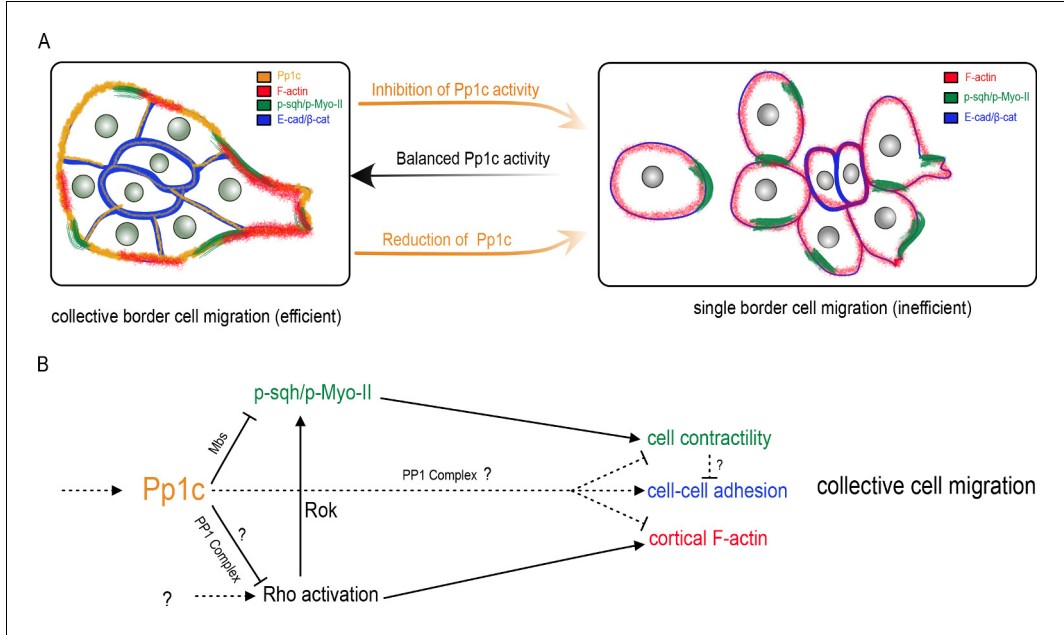

**Figure 8.** Model for the Pp1 function in border cell migration. (**A**) Schematic of the phenotypes and the localizations of Pp1c, F-actin, p-Sqh, and the cadherin-catenin complex during normal and Pp1-inhibited (NiPp1 expression or *Pp1c-RNAi*) border cell cluster migration. (**B**) Proposed molecular pathways regulated by Pp1, which together promote cohesive collective border cell migration.

proteins in *Drosophila*, several of which have domains associated with activation by Rho GTPases. However, which formin, if any, promotes border cell migration and F-actin distribution is unknown. Further work will be needed to determine whether any of these potential targets, or other actin regulatory proteins, control collective level F-actin enrichment via Pp1.

A major consequence of decreased Pp1 activity is fragmentation of the border cell cluster into single border cells and small groups. This raises the question of how Pp1 activity maintains cluster cohesion, which is critical for collective cell movement in vivo. Like many cell collectives, high levels of cadherin-catenin complex proteins are detected between all border cells (*Niewiadomska et al., 1999*). The cadherin-catenin complex is required for border cells to adhere to the central polar cells as well as to provide migratory traction of the entire cluster upon the nurse cells (*Cai et al., 2014*; *Niewiadomska et al., 1999*). We found that Pp1 maintains E-Cadherin and β-Catenin levels between border cells. Indeed, other mutants that disrupt the levels and localization of adhesion proteins in border cells often also disrupt cluster shape and cohesion. For example, loss of JNK signaling causes border cell clusters to dramatically elongate, with downregulation of adhesion resulting in incomplete separation of border cells (*Llense and Martín-Blanco, 2008*; *Melani et al., 2008*). Raskol, a putative Ras guanine nucleotide activating protein (GAP), maintains E-cadherin at BC-BC contacts and cohesion of the cluster (*Raza et al., 2019*). However, while loss of *Raskol* causes a significant number of border cells to fully dissociate from the cluster (~35%) (*Raza et al., 2019*), similar to what we observe with knockdown of the cadherin-catenin complex, this is less than what we observe upon inhibition of Pp1 activity (~90%). Thus, while cluster fragmentation caused by Pp1 inhibition is at least partly due to deficient cadherin-catenin adhesion, other targets likely contribute.

Our results indicate that E-Cadherin, β-Catenin, and α-Catenin maintain adhesion of border cells to each other in addition to known roles in keeping border cells attached to the polar cells (*Cai et al., 2014*). Knockdown of the cadherin-catenin complex members in both border cells and polar cells causes border cells to significantly dissociate from the cluster. The requirement in border cells for cadherin-catenin in cluster cohesion may have been masked in prior studies due to the inability of strong loss-of-function cadherin-catenin mutant border cells to move at all (*Cai et al., 2014*; *Niewiadomska et al., 1999*; *Sarpal et al., 2012*; *Desai et al., 2013*). While RNAi for E-Cadherin, β-Catenin, and α-Catenin each strongly knock down the respective protein levels, it may be

that a small amount of each protein is still present. Such remaining cadherin-catenin proteins may provide just enough traction for border cells to partially migrate upon the nurse cells. We speculate that movement of cadherin-catenin-deficient border cells within the confining tissue would provide mechanical stresses that break the cluster apart at weakened border cell-border cell contacts. Indeed, a mutant α-Catenin protein that lacks part of the C-terminal F-actin-binding domain was shown to partially rescue the migration defects caused by loss of *α-Catenin*; however, these rescued border cell clusters split into several parts along the migration path (*Desai et al., 2013*). Further supporting this idea, Pp1-inhibited border cells fall apart during their effort to migrate between the nurse cells.

How do Pp1 phosphatase complexes molecularly promote cluster cohesion? Given the effects of Pp1 on E-Cadherin and β-Catenin at internal border cell contacts, and the requirement for cadherin-catenin complex proteins in maintaining cluster integrity, Pp1 could directly regulate cadherin-catenin protein stability and/or adhesive strength. In mammalian and *Drosophila* cells, phosphorylation of a conserved stretch of serine residues in the E-Cadherin C-terminal tail region regulates E-Cadherin protein stability, binding of E-Cadherin to β-Catenin, and cell-cell junction formation and turnover (*Stappert and Kemler, 1994*; *McEwen et al., 2014*; *Chen et al., 2017*). Serine-phosphorylation of α-Catenin is also required for adhesion between epithelial cells and possibly for efficient border cell migration (*Escobar et al., 2015*). More work will be needed to determine whether a to-be-identified Pp1-containing phosphatase complex directly dephosphorylates E-Cadherin and/or α-Catenin, as the roles for phosphatases in cadherin-catenin junctional stability are still poorly understood.

Alternatively, or in addition, Pp1-dependent restriction of collective actomyosin contraction to the cluster periphery could allow internal cluster cell-cell junctions to be maintained. Pp1-inhibition greatly alters actomyosin distribution, causing individual border cells to contract and round up. The forces transmitted by high cell contractility alone could weaken adherens junctions, causing the border cells to break apart during migration (*Figure 8A*). Myosin phosphatase-depleted border cells, which have elevated phosphorylated Sqh (*Majumder et al., 2012*), and thus active Myo-II, are round, highly contractile, and fall off the cluster. In support of this idea, overexpression of a phosphorylation mutant form of Sqh (Sqh$^{E20E21}$), which mimics activated Myo-II, causes border cells to have a similar round shape and separate from the cluster (*Mishra et al., 2019*). Thus, collective-level active actomyosin contraction contributes to keeping border cells adhered to the cluster. Myo-II and cadherin-catenin complexes have dynamic and quite complex interactions that influence stability of cell-cell junctions, and which may depend on cellular context (*Mège and Ishiyama, 2017*; *Yap et al., 2018*). In border cells, the cadherin-catenin complex promotes enrichment of actomyosin to the cluster periphery, whereas Myo-II does not greatly influence cadherin-catenin levels within the cluster (this study) (*Mishra et al., 2019*). However, Pp1 is required for the proper distribution (or stability) of cadherin-catenin at cell contacts between border cells and prevents the enrichment of actomyosin in individual border cells. Moreover, NiPp1 expression disrupts cluster cohesion to a greater extent than knockdown of either myosin phosphatase or cadherin-catenin complex members alone. This suggests that cadherin-catenin stability and optimal collective-wide actomyosin activity both contribute to cluster cohesion through distinct Pp1 phosphatase complexes, although this possibility remains to be formally tested (*Figure 8B*).

Our study implicates Pp1 as a major regulator of collective cohesion and migration in border cells. Pp1 catalytic subunits and their regulatory subunits are conserved across eukaryotes (*Lin et al., 1999*; *Verbinnen et al., 2017*; *Heroes et al., 2013*; *Ferreira et al., 2019*). The roles of specific Pp1 complexes in collective cell migration during development and in cancer have not been well studied. Intriguingly, Mypt1 (Mbs homolog) promotes polarized mesodermal migration during zebrafish gastrulation (*Weiser et al., 2009*). Similar to what we observe in Mbs-depleted border cells, inhibition of zebrafish Mypt1 switched cells from an elongated mesenchymal mode of migration to a hyper-contractile amoeboid mode of migration. Another Pp1 phosphatase complex containing the Phactr4 (phosphatase and actin regulator 4) regulatory subunit promotes the chain-like collective migration of enteric neural crest cells, which colonize the gut and form the enteric nervous system during development (*Zhang et al., 2012*). Phactr4, through Pp1, specifically controls the directed migration and shape of enteric neural crest cells through integrin, Rok, and cofilin. Given the conservation of these and other phosphatase complexes, our study highlights the importance of balanced Pp1 phosphatase activity in the organization and coordination of migrating cell collectives.

# Materials and methods

## Key resources table

| Reagent type (species) or resource | Designation | Source or reference | Identifiers | Additional information |
|---|---|---|---|---|
| Genetic reagent (*Drosophila melanogaster*) | *c306*-GAL4 tsGAL80 | *Aranjuez et al., 2016* | | Laboratory of Jocelyn McDonald |
| Genetic reagent (*D. melanogaster*) | *slbo*-GAL4 | other | FBal0089668 | from D. Montell |
| Genetic reagent (*D. melanogaster*) | *upd*-GAL4 | other | FBal0047063 | from D. Montell |
| Genetic reagent (*D. melanogaster*) | *c306*-GAL4 | Bloomington *Drosophila* Stock Center | BDSC Cat# 3743; RRID:BDSC_3743 | |
| Genetic reagent (*D. melanogaster*) | *UAS*-NiPp1.HA | Bloomington *Drosophila* Stock Center | BDSC Cat# 23711; RRID:BDSC_23711 | |
| Genetic reagent (*D. melanogaster*) | *UAS*-Pp1-87B.HA | Bloomington *Drosophila* Stock Center | BDSC Cat# 24098; RRID:BDSC_24098 | |
| Genetic reagent (*D. melanogaster*) | *UAS*-Pp1-13C.HA | Bloomington *Drosophila* Stock Center | BDSC Cat# 23701; RRID:BDSC_23701 | |
| Genetic reagent (*D. melanogaster*) | *UAS*-Pp1alpha-96A.HA | Bloomington *Drosophila* Stock Center | BDSC Cat# 23700; RRID:BDSC_23700 | |
| Genetic reagent (*D. melanogaster*) | *UAS*-hPPP1CC | Bloomington *Drosophila* Stock Center | BDSC Cat# 64394; RRID:BDSC_64394 | |
| Genetic reagent (*D. melanogaster*) | *UAS*-mCherry RNAi | Bloomington *Drosophila* Stock Center | BDSC Cat# 35785; RRID:BDSC_35785 | VALIUM20-mCherry |
| Genetic reagent (*D. melanogaster*) | *UAS*-mCD8.ChRFP | Bloomington *Drosophila* Stock Center | BDSC Cat# 27392; RRID:BDSC_27392 | |
| Genetic reagent (*D. melanogaster*) | *flw*$^{FP41}$ FRT 19A | Bloomington *Drosophila* Stock Center | BDSC Cat# 51338; RRID:BDSC_51338 | |
| Genetic reagent (*D. melanogaster*) | *Ubi*-mRFP.nls, hsFLP, FRT19A | Bloomington *Drosophila* Stock Center | BDSC Cat# 31418; RRID:BDSC_31418 | |
| Genetic reagent (*D. melanogaster*) | *UAS*-PLCδ-PH-GFP | Bloomington *Drosophila* Stock Center | BDSC Cat# 39693; RRID:BDSC_39693 | |
| Genetic reagent (*D. melanogaster*) | *UAS*-Pp1α−96A RNAi | Vienna *Drosophila* Resource Center | VDRC:27673 | GD-11970 |
| Genetic reagent (*D. melanogaster*) | *UAS*-Pp1-87B RNAi | Vienna *Drosophila* Resource Center | VDRC:35024 | GD-11720 |
| Genetic reagent (*D. melanogaster*) | *UAS*-Pp1-13C RNAi | Vienna *Drosophila* Resource Center | VDRC:29057 | GD-14139 |
| Genetic reagent (*D. melanogaster*) | *UAS*-Mbs RNAi | Vienna *Drosophila* Resource Center | VDRC:105762 | KK-109231 |
| Genetic reagent (*D. melanogaster*) | *UAS*-E-cad RNAi | Vienna *Drosophila* Resource Center | VDRC:103962 | KK-103334 |
| Genetic reagent (*D. melanogaster*) | *UAS*-E-cad RNAi | Vienna *Drosophila* Resource Center | VDRC:27082 | GD-14421 |
| Genetic reagent (*D. melanogaster*) | *UAS*-β-cat RNAi | Vienna *Drosophila* Resource Center | VDRC:107344 | KK-102545 |
| Genetic reagent (*D. melanogaster*) | *UAS*-β-cat RNAi | Vienna *Drosophila* Resource Center | BDSC:31305 | TRiP.JF01252 |
| Genetic reagent (*D. melanogaster*) | *UAS*-α-cat RNAi | Vienna *Drosophila* Resource Center | VDRC:107298 | KK-107916 |
| Genetic reagent (*D. melanogaster*) | *UAS*-α-cat RNAi | Vienna *Drosophila* Resource Center | VDRC:20123 | GD-8808 |
| Genetic reagent (*D. melanogaster*) | *fTRG sqh* | Vienna *Drosophila* Resource Center | VDRC:318484 | fTRG 10075 |

*Continued on next page*

*Continued*

| Reagent type (species) or resource | Designation | Source or reference | Identifiers | Additional information |
|---|---|---|---|---|
| Genetic reagent (*D. melanogaster*) | *fTRG Pp1α −96A* | Vienna *Drosophila* Resource Center | VDRC:318084 | fTRG 290 |
| Genetic reagent (*D. melanogaster*) | flwCPTI002264 | Kyoto *Drosophila* Genomics and Genetic Resources | line 115284 | FBti0143758 |
| Genetic reagent (*D. melanogaster*) | *UAS*-Flw.HA | The Zurich ORFeome Project,FlyORF | line F001200 | |
| Antibody | rat monoclonal anti-E-cadherin | Developmental Studies Hybridoma Bank | DCAD2; RRID:AB_528120 | 1:10 |
| Antibody | mouse monoclonal anti-Fasciclin III | Developmental Studies Hybridoma Bank | 7G10; RRID:AB_528238 | 1:10 |
| Antibody | mouse monoclonal anti-Arm | Developmental Studies Hybridoma Bank | N2-7A1; RRID:AB_528089 | 1:75 |
| Antibody | mouse monoclonal anti-Fascin (Singed) | Developmental Studies Hybridoma Bank | sn 7C; RRID:AB_528239 | 1:25 |
| Antibody | rabbit polyclonal anti- Phospho-Myosin Light Chain 2 (Ser19) | Cell Signaling Technology, Inc | #3671; RRID:AB_330248 | 1:10 |
| Antibody | rat monoclonal anti-HA (3F10) | Millipore Sigma | 11867423001; RRID:AB_2314622 | 1:1000 |
| Antibody | rabbit polyclonal anti-Mbs | *Ong et al., 2010* | | 1:200 from Change Tan |
| Antibody | rabbit polyclonal anti-GFP | Thermo Fisher Scientific | A11122; RRID:AB_221569 | 1:1000–1:2000 |
| Antibody | chicken polyclonal anti-GFP | Abcam | ab13970; RRID:AB_300798 | 1:1000 |
| Antibody | rabbit polyclonal anti-PPP1R8 (NiPP1) | Millipore Sigma | HPA027452; RRID:AB_1854490 | 1:100 |
| Antibody | Alexa Fluor 488, 568, or 647 | Thermo Fisher Scientific | | 1:400 |
| Chemical compound, drug | Alexa Fluor 488 or 568 Phalloidin | Thermo Fisher Scientific | A12379 or A12380 | 1:400 |
| Chemical compound, drug | Phalloidin-Atto 647N | Millipore Sigma | 65906 | 1:400 |
| Chemical compound, drug | 4′,6-Diamidino-2-phenylindole (DAPI) | Millipore Sigma | D9542 | 0.05 µg/ml |
| Software, algorithm | FIJI | PMID:22743772 | | |
| Software, algorithm | Graphpad Prism 7, Prism 8 | https://www.graphpad.com/ | | |
| Software, algorithm | Adobe Photoshop CC | https://www.adobe.com/ | | |
| Software, algorithm | Adobe Illustrator CC 2018 | https://www.adobe.com/ | | |
| Software, algorithm | Affinity Designer 1.7.1 | https://affinity.serif.com/ | | |
| Software, algorithm | Zeiss AxioVision 4.8 | Zeiss | | |
| Software, algorithm | Zeiss ZEN 3.0 | Zeiss | | |
| Software, algorithm | Final Cut Pro X 10.4.8 | Apple | | |

## *Drosophila* genetics and strains

Crosses were generally set up at 25°C unless otherwise indicated. The *tub*-GAL80^ts ('tsGAL80') transgene (*McGuire et al., 2004*) was included in many crosses to suppress GAL4-UAS expression during earlier stages of development; these crosses were set up at 18°−22 °C to turn on tsGAL80. For *c306*-GAL4, *c306*-GAL4-tsGal80, *slbo*-GAL4, or *upd*-GAL4 tsGAL80 crosses, flies were incubated at 29°C for ≥14 hr prior to dissection to produce optimal GAL4-UAS transgene expression. *c306*-GAL4 is expressed early and more broadly in border cells, polar cells, and terminal (anterior and posterior) follicle cells (*Figure 1—figure supplement 1A*; *Figure 1—figure supplement 2B*; *Silver and Montell, 2001*). During oogenesis, *slbo*-GAL4 turns on later than *c306*-GAL4, and is expressed in border cells but not polar cells, as well as a few anterior and posterior follicle cells at stage 9 (*Figure 1—figure supplement 1B*; *Figure 1—figure supplement 2C,D*; *Silver and Montell, 2001*; *Rørth et al., 1998*). *upd*-GAL4 is restricted to polar cells at all stages of oogenesis (*Figure 1—figure supplement 2C,H*; *Cai et al., 2014*). Mosaic mutant clones of *flw* were generated using the FLP-FRT system (*Xu and Rubin, 1993*). The *flw^FP41* FRT 19A line was crossed to *ubi*-mRFP.nls *hs*FLP FRT19A; the resulting progeny were heat shocked for 1 hr at 37°C, two times a day for 3 d, followed by 3 d at 25°C prior to fattening and dissection. Mutant clones were identified by loss of nuclear RFP signal from *ubi*-mRFP.nls.

The following *Drosophila* strains (with indicated stock numbers) were obtained from the Bloomington *Drosophila* Stock Center (BDSC, Bloomington, IN, USA): *c306*-GAL4 (3743), UAS-NiPp1.HA (23711), UAS-Pp1-87B.HA (24098), UAS-Pp1-13C.HA (23701), UAS-Pp1α−96A.HA (23700), UAS-hPPP1CC (64394), UAS-mCD8-ChRFP (27392), UAS-mCherry RNAi (35785), UAS-Pp2B-14D RNAi (25929, 40872), UAS-mts RNAi (27723, 38337, 57034, 60342), UAS-Pp4-19C RNAi (27726, 38372, 57823), UAS-CanA-14F RNAi (38966), UAS-PpD3 RNAi (57307), UAS-PpV RNAi (57765), UAS-CanA1 RNAi (25850), UAS-CG11597 RNAi (57047, 61988), UAS-rgdC RNAi (60076), UAS-Flw RNAi (38336), UAS-β-Catenin RNAi JF01252 (31305), *flw^FP41* FRT 19A (51338), ubi-mRFP.nls hsFLP FRT19A (31418), UAS-PLCδ-PH-EGFP ('membrane GFP'; 39693), UAS-GFP.nls (4776).

The following *Drosophila* strains (with indicated stock numbers) were obtained from the Vienna *Drosophila* Resource Center (VDRC, Vienna, Austria): UAS-Pp1α−96A RNAi (v27673), UAS-Pp1-87B RNAi (v35024), UAS-Pp1-13C RNAi (v29058), UAS-Flw RNAi (v29622, v104677), UAS-Mbs RNAi (v105762), UAS-Pp2B-14D RNAi (v46873), UAS-Pp4-19c RNAi (25317), UAS-E-Cadherin RNAi (v27082, v103962), UAS-β-Catenin RNAi (v107344), UAS-α-Catenin RNAi (v20123, v107298), UAS-Sqh RNAi (v7916), fTRG Pp1α −96A (v318084), fTRG Sqh (v318484).

Other *Drosophila* strains used in this study were: *slbo*-GAL4, *slbo*-GAL4 UAS-mCD8-GFP, *upd*-GAL4;; tsGAL80, and *slbo*-LifeAct-GFP line 2M (from D. Montell, University of California, Santa Barbara, Santa Barbara, CA, USA), *flw^CPTI002264* protein trap (line 115284, Kyoto Stock Center, Kyoto, Japan), UAS-mCherry-Jupiter (from C. Doe, University of Oregon, Eugene, OR, USA), UAS-Rac FRET (*Wang et al., 2010*), UAS-Rho FRET/CyO; UAS-Rho FRET/TM6B (*Qin et al., 2017*), and UAS-Flw.HA (FlyORF) (*Bischof et al., 2013*). The *c306*-GAL4 tsGAL80 (*Aranjuez et al., 2016*) and *c306*-GAL4 tsGAL80/FM6; UAS-NiPp1.HA/TM3 Ser stocks were created in our lab.

## Female fertility test

Fertility was determined according to established methods (*Tootle and Spradling, 2008*). Briefly, four *c306*-GAL4 tsGAL80/FM6; Sco/CyO (control) or *c306*-GAL4 tsGAL80/FM6; UAS-NiPP1/TM3 Ser (experimental) females were outcrossed to four *w^1118* males. The flies were allowed to mate for 2 days followed by a 24 hr egg lay at 30 °C on fresh food medium supplemented with yeast. Adults were then removed and the progeny allowed to develop in the vial at 25 °C; the food was periodically monitored to avoid drying out. Scoring of eclosed adult progeny from each vial was performed 16–20 d after egg laying and reported as the average progeny per female.

## Immunostaining

Fly ovaries from 3- to 5-d-old females were dissected in Schneider's *Drosophila* Medium (Thermo Fisher Scientific, Waltham, MA, USA) supplemented with 10% fetal bovine serum (Seradigm FBS; VWR, Radnor, PA, USA). Ovaries were kept whole or dissected into individual egg chambers, followed by fixation for 10 min using 4% methanol-free formaldehyde (Polysciences, Warrington, PA, USA) in 0.1 M potassium phosphate buffer, pH 7.4, or in 1X Phosphate Buffered Saline (PBS).

Washes and antibody incubations were performed in 'NP40 block' (50 mM Tris-HCl, pH 7.4, 150 mM NaCl, 0.5% NP40, 5 mg/ml bovine serum albumin [BSA]). For α-Catenin immunostaining, dissected egg chambers were fixed for 20 min in 4% paraformaldehyde (Electron Microscopy Sciences, Hatfield, PA, USA) in potassium phosphate buffer, pH 7.4, followed by a separate blocking step for 30 min (2% BSA in 1x PBS) prior to each antibody incubation. For p-Sqh antibody staining, ovaries were fixed for 5 min in 8% methanol-free formaldehyde. For the F-actin staining in *Figure 6*, the entire dissection procedure was performed in less than 10 min to preserve F-actin structures, followed by fixation in the presence of Phalloidin at 1:400 dilution; after washing off the fix, the egg chambers were incubated in Phalloidin at 1:400 for 2 h (*Spracklen et al., 2014*).

The following primary antibodies from the Developmental Studies Hybridoma Bank (DSHB, University of Iowa, Iowa City, IA, USA) were used at the indicated concentrations: rat anti-E-Cadherin 1:10 (DCAD2), mouse anti-Fasciclin III 1:10 (FasIII; 7G10), mouse anti-Arm (β-Catenin) 1:75 (N2-7A1), concentrated rat anti-α-Catenin 1:1000 (DCAT1), mouse anti-Eyes Absent 1:100 (eya10H6), mouse anti-Lamin Dm0 1:10 (ADL67.10), and mouse anti-Singed 1:25 (Sn7C). Additional primary antibodies used were: rabbit anti-Phospho-Myosin Light Chain 2 (Ser19) 1:10 (#3671, Cell Science Technology, Danvers, MA, USA), rat anti-HA 1:1000 (11867423001, Millipore Sigma, Burlington, MA, USA), rabbit anti-Mbs 1:200 (from C. Tan, University of Missouri, Columbia, MO, USA); rabbit anti-GFP polyclonal 1:1000-1:2000 (A-11122, Thermo Fisher Scientific), chicken anti-GFP polyclonal 1:1000 (ab13970, Abcam, Cambridge, MA, USA), rabbit anti-PPP1R8 (NiPP1) polyclonal 1:100 (HPA027452, Millipore Sigma), rat anti-Slbo 1:2000 (from P. Rørth, Institute of Molecular and Cell Biology, Singapore). Alexa Fluor 488, 568, or 647 secondary antibodies (Thermo Fisher Scientific) were used at 1:400 dilution. Alexa Fluor Phalloidin (488 or 568; Thermo Fisher Scientific) and Phalloidin–Atto 647N (Millipore Sigma) were used at 1:400 dilution. 4',6-Diamidino-2-phenylindole (DAPI, Millipore Sigma) was used at 0.05 µg/ml. Egg chambers were mounted on slides with Aqua-Poly/Mount (Polysciences) or FluorSave Reagent (Millipore Sigma) for imaging.

## Microscopy, live time-lapse imaging, and FRET

Images of fixed egg chambers were acquired with an upright Zeiss AxioImager Z1 microscope and Apotome.2 optical sectioning, or on a Zeiss LSM 880 confocal microscope with or without Airyscan (KSU College of Veterinary Medicine Confocal Core), using either a 20 × 0.75 numerical aperture (NA) or 40 × 1.3 NA oil-immersion objective.

Live time-lapse imaging was performed as described (*Prasad and Montell, 2007*; *Dai and Montell, 2016*). Briefly, ovarioles were dissected in room-temperature sterile live imaging media (Schneider's *Drosophila* Medium, pH 6.95, with 15–20% FBS). Fresh live imaging media, supplemented with 0.2 µg/ml bovine insulin (Cell Applications, San Diego, CA, USA), was added to the sample prior to mounting on a lumox dish 50 (94.6077.410; Sarstedt, Newton, NC, USA). Time-lapse videos were generally acquired at intervals of 2–3 min for 3–6 hr using a 20 × Plan Apochromat 0.75 NA objective, a Zeiss Colibri LED light source, and a Zeiss Axiocam 503 mono camera. The LED light intensity was experimentally adjusted to maximize fluorescence signal and to minimize phototoxicity of the live sample. Live time-lapse Sqh-GFP imaging was performed on a Zeiss LSM 880 confocal, as described (*Dai and Montell, 2016*), with a 40 × 1.2 NA water-immersion objective using an interval of 1 min for up to 20 min total time and a laser setting of 1.5%. Imaging gain and other acquisition parameters were the same, except that the range of z-stacks varied slightly depending on the sample. In some cases, multiple z-stacks were acquired and merged in Zeiss AxioVision, Zeiss ZEN 2, or FIJI (*Schindelin et al., 2012*) to produce a single, in-focus time-lapse video.

FRET images (Rac FRET, Rho FRET) of live cultured egg chambers were acquired with a Zeiss LSM710 microscope essentially as described (*Wang et al., 2010*). A 40 × 1.3 NA oil inverted objective was used to capture single high-resolution stationary images. A 458 nm laser was used to excite the sample. CFP and YFP emission signals were collected through channel I (470–510 nm) and channel II (525–600 nm), respectively. The CFP and YFP channels were acquired simultaneously for most experiments. Sequential acquisition of CFP and YFP channels was tested but produced the same result as simultaneous acquisition.

## Image processing and data analysis

Image measurements and editing were performed using Zeiss ZEN or FIJI (*Schindelin et al., 2012*). Analyses of live border cell migration time-lapse videos was performed using Zeiss ZEN software. The migration speed was calculated from the duration of border cell movement. Protrusion quantification was performed as described (*Sawant et al., 2018*). Briefly, a circle was drawn around the cell cluster, and extensions greater than 1.5 μm outside the circle were defined as protrusions (*Figure 5—figure supplement 1A*). Protrusions were classified as directed to the front (0˚−45˚ and 0˚−315˚), side (45˚−135˚and 225˚−315˚), or back (135˚−225˚), based on their positions within the cluster. The first 1 hr of each video was used for protrusion quantification.

To determine the number of cells per cluster, egg chambers were stained for the nuclear envelope marker Lamin, the DNA stain DAPI, and the cell membrane marker E-Cadherin. Only clusters that had delaminated, moved forward, and had any detectable E-Cadherin were imaged. This allowed confidence that the scored cells were border cells. Acquisition of z-stacks that encompassed the entire cluster (border cells and polar cells) were defined by nuclear Lamin signal. This was followed by manual counting of the nuclei from the resulting images.

The circularity of border cells was measured in FIJI. Individual border cells were outlined manually based on the PLCδ-PH-GFP signal using the 'Freehand Selections' tool. Within the 'Set Measurements' analysis tool, 'shape descriptors' was selected, followed by the 'Measure' function, which provided a measurement of circularity. A value of 1.0 indicates a perfect circle, whereas 0.0 represents an extremely elongated shape.

Measurements of E-Cadherin and β-Catenin intensity at cell–cell junctions were performed on egg chambers that were stained using identical conditions. Samples were imaged with a 40 × 1.3 NA oil objective. Identical confocal laser settings were used for each channel and a full z-stack of the cluster was produced. Images were then subjected to 3D reconstruction through the '3D Project' function in FIJI. Border cell-border cell (BC-BC) contacts and nurse cell-nurse cell (NC-NC) contacts were manually identified, a line (width set as 6) drawn, and mean fluorescence intensity across the line was obtained using the 'measure' tool. A ratio of BC-BC intensity versus NC-NC intensity was calculated to normalize protein levels.

To measure colocalization between Mbs and Flw, or Mbs and Pp1α−96A, the 'RGB Profiler' FIJI plugin was used. After converting the image to RGB, a line was drawn across the whole border cell cluster to generate the image intensity plot. The localization patterns of F-actin and Mbs with Pp1α−96A-GFP and Flw-YFP were measured through the 'Analyze >Plot Profile' function in FIJI. A line was drawn across the border cells and polar cells and the pixel intensity value was obtained across the line. The values for each channel were normalized to the highest pixel value, and a scatter plot showing F-actin and DAPI was generated in Microsoft Excel.

For Rho-FRET and Rac-FRET, the CFP and YFP images were first processed in ImageJ. A background region of interest was subtracted from the original image. The YFP images were registered to CFP images using the TurboReg plugin. The Gaussian smooth filter was then applied to both channels. The YFP image was thresholded and converted to a binary mask with the background set to zero. The final ratio image was generated in MATLAB, during which only the unmasked pixels were calculated as described (*Wang et al., 2010*).

## Figures, graphs, and statistics

Figures were assembled in Adobe Photoshop CC. Illustrations were created in Affinity Designer (Serif, Nottingham, United Kingdom). Videos were assembled in Zeiss AxioVision 4.8, Zeiss ZEN 2, or FIJI. Graphs and statistical tests were performed using GraphPad Prism 7 or Prism 8 (GraphPad Software, San Diego, CA, USA). The statistical tests and *p* values are listed in the figure legends.

## Acknowledgements

We thank Drs. Chris Doe, Denise Montell, Pernille Rørth, Change Tan, Tina Tootle, Alan Zhu, the Bloomington *Drosophila* Stock Center, the Developmental Studies Hybridoma Center (University of Iowa), FlyORF, the Kyoto Stock Center, and Vienna *Drosophila* Resource Center (VDRC) for fly stocks, antibodies, and protocols. We also thank Dr. Pralay Majumder for discussions and providing comments on the manuscript and Kristen Hylen for making the initial observations with α-Catenin

RNAi. The Confocal Core, funded by the Kansas State University College of Veterinary Medicine, provided use of the Zeiss LSM 880 confocal microscope. This work was supported in part by a fellowship from the Kansas INBRE through the National Institutes of Health (P20 GM103418) to YC, and by grants from the Scientifiques de la Fondation ARC (grant number PJA 20191209714) to XW and the National Science Foundation (1456053 and 1738757) to JAM.

## Additional information

### Funding

| Funder | Grant reference number | Author |
| --- | --- | --- |
| National Institutes of Health | P20 GM103418 | Yujun Chen |
| ARC Foundation for Cancer Research | PJA20191209714 | Xiaobo Wang |
| National Science Foundation | 1456053 | Jocelyn A McDonald |
| National Science Foundation | 1738757 | Jocelyn A McDonald |

The funders had no role in study design, data collection and interpretation, or the decision to submit the work for publication.

### Author contributions

Yujun Chen, Conceptualization, Formal analysis, Funding acquisition, Validation, Investigation, Visualization, Methodology; Nirupama Kotian, George Aranjuez, Lin Chen, C Luke Messer, Ashley Burtscher, Ketki Sawant, Formal analysis, Validation, Investigation, Visualization, Methodology; Damien Ramel, Conceptualization, Formal analysis, Validation, Investigation, Visualization, Methodology; Xiaobo Wang, Conceptualization, Resources, Formal analysis, Supervision, Funding acquisition; Jocelyn A McDonald, Conceptualization, Formal analysis, Supervision, Funding acquisition, Validation, Investigation, Visualization, Methodology

### Author ORCIDs

Yujun Chen (iD) http://orcid.org/0000-0003-3190-6713
Damien Ramel (iD) https://orcid.org/0000-0002-9089-1497
Jocelyn A McDonald (iD) https://orcid.org/0000-0002-7494-1466

### Decision letter and Author response

Decision letter https://doi.org/10.7554/eLife.52979.sa1
Author response https://doi.org/10.7554/eLife.52979.sa2

## Additional files

### Supplementary files

• Transparent reporting form

### Data availability

All data generated or analysed during this study are included in the manuscript and supporting files.

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
