## [Decision Letter]

**Acceptance summary:**

How groups of cells migrate directionally as cohesive clusters is still not well understood. In this paper the authors demonstrate that in the fly ovary the Pp1 phosphatase plays a crucial role in regulating cell delamination, migration and cohesion of polar cells. The authors performed a detailed analysis and quantification of the phenotypes resulting from knockdown of different subunits. The resulting phenotypes are quite striking with the border cell collective falling apart, rounding of the cells and frequent failure to reach the oocyte. The paper shows that via different subunits Pp1 controls adherence of the cells to the cluster as well as cell shape, thereby regulating collective cell versus single cell behaviors.

**Decision letter after peer review:**

Thank you for submitting your work "Protein Phosphatase 1 activity controls a balance between collective and single cell modes of migration". Your article has been reviewed by three reviewers, one of whom is a member of our Board of Reviewing Editors, and the evaluation has been overseen by Utpal Banerjee as the Senior Editor.

The reviewers have discussed the reviews with one another and the Reviewing Editor has drafted this decision to help you prepare a revised submission.

Summary:

In this manuscript by Chen et al., the authors discovered that protein phosphatase 1 (Pp1) globally controls the organization, cohesion, and coordination of *Drosophila* border cells. Specifically, they show that Pp1 activity, mediated through distinct phosphatase complexes such as myosin phosphatase is as a critical regulator of collective cell versus single cell behaviors. To test possible functions of phosphatases in border cell migration the authors screened selected serine-threonine phosphatases that are expressed during oogenesis via RNAi or the inhibitor NiPp1. The authors performed a detailed analysis and quantification of the phenotypes resulting from knockdown of different subunits. The resulting phenotypes are quite striking with the border cell collective falling apart, rounding of the cells and frequent failure to reach the oocyte. Using a candidate gene approach the authors show that cadherin-catenin expression is downregulated and that loss of cadherin-catenin complex members causes similar phenotypes as loss of Pp1. In addition, rounding of cell shapes suggested alterations in the cortical cytoskeleton, which is supported by loss of polarized F-actin and Myo-II.

While the manuscript is interesting, beautifully illustrated and accompanied by a scholarly written discussion, the reviewers feel that some additional experiments will lead to a better mechanistic understanding of the function of Pp1.

1) The temporal dynamics of the phenotypes needs to be described in more detail. Specifically, the authors should clarify phenotypes in reference to the different stages of migration and investigate if the different Pp1 catalytic subunits possess distinct, stage-specific roles (e.g. delamination vs. active migration) or if all of their functions are overlapping.

2) In addition, the authors conclude that Pp1 globally coordinates the collective behaviors of border cells, from adhesion of cells within the cluster, to formation of migratory protrusions, formation of proper cell shapes, and efficient group motility. From the study it is unclear if these phenotypes are directly controlled by Pp1 or if some of them could be, for example, secondary to the loss of adhesion or the mis-localization of Myo-II. The authors mention in the Discussion section that Myo-II and cadherin-catenin complexes possess dynamic interactions that influence stability of cell-cell junctions, suggesting that the adhesion defect could be secondary to the defect in Myo-II distribution.

It would be helpful to already mention and test such possible interactions in the result section. The authors should determine how F-actin, Myo-II and cadherin-catenin complexes are localized in the respective RNAi experiments.

3) The results pertaining to p-Sqh/Myosin and *Mbs*-RNAi, in particular, are less convincing and need to be strengthened (Figure 7) as the phenotypes were not very clear (e.g. Video 14). Likewise, Video 16 does not convincingly support the conclusion that sqh:GFP is more uniform, “especially at contacts between some border cells”.

“Pp1c border cells followed the normal migratory path down the center of the egg chamber between nurse cells, even if when cells broke off…” This does not seem to be the case in Video 7. Please, also indicate how many videos were obtained for each experiment.

4) Furthermore, while this study does a nice job of putting a lot of data into context, some results need to be more clearly explained in the context of the field, to clarify new results from those that have been previously documented- particularly with respect to the previous results of the Mbs subunit of PP1 (Aranjuez et al., 2016) and if such drastic loss of border cell cohesion has been previously observed after any other experimental manipulations.

---

## [Author Response]

1) The temporal dynamics of the phenotypes needs to be described in more detail. Specifically, the authors should clarify phenotypes in reference to the different stages of migration and investigate if the different Pp1 catalytic subunits possess distinct, stage-specific roles (e.g. delamination vs. active migration) or if all of their functions are overlapping.

We agree that this is an important point. Delamination (also called “detachment”) from the follicular epithelium is an essential step in border cell collective migration, as it controls movement of the cohesive cluster out of the epithelium and into the egg chamber proper. We have now re-analyzed the *Pp1* mutant (RNAi) phenotypes from our previously acquired data, both in fixed samples and in live time-lapse imaging, which is now shown in Figure 3E and Figure 3—figure supplement 1A.

In fixed samples, we asked if the border cells delaminated from the follicular epithelium and partially migrated (delaminated, but stopped anywhere along the migration pathway), delaminated and reached the oocyte (“complete” migration), or did not delaminate/migrate (“no migration”). The majority of Pp1c-inhibited border cells delaminated but stopped along the migration pathway. However, in the case of *Pp1α-96A-*RNAi border cells, ~15% of border cells did not migrate (*p* < 0.001) suggesting that at least this Pp1c subunit has a role in delamination (Figure 3E). *flw* mosaic clones also have split border cells that do not migrate (Figure 3J). The other two Pp1c subunits had a few border cells fail to delaminate/migrate, but this was not statistically significant (Figure 3E).

To confirm that *Pp1c* is required for delamination, we analyzed the live time-lapse videos for delamination by the end of imaging. Because these clusters often split (e.g. Figure 3B-D,F), we analyzed each individual “part”, or small group of dissociated border cells, within each video. As with the fixed samples, *Pp1α-96A-*RNAi live border cells had a significant fraction of border cells that did not delaminate during the video, compared to *Pp1-87B* or *Pp1-13C* RNAi in which most border cells delaminated (Figure 3—figure supplement 1A). This is also supported by Video 8, where some *Pp1α-96A-*RNAi border cells visibly had trouble delaminating and did not leave the epithelium during imaging. These data are now described in the text subsection “Pp1c genes are required for border cell cluster migration and cohesion”.

Both *Pp1α-96A*-RNAi and *flw* mosaic clones result in more delamination defects (no migration) than that observed with knockdown of the other two *Pp1c* subunits. These results suggest roles for Flw and Pp1α-96A during delamination, with all Pp1c subunits required to keep border cells together and efficiently migrate once the clusters delaminate. However, due to potential partial knockdown by RNAi we cannot completely rule out overlapping roles for the other two Pp1c subunits during delamination. Moreover, we observed full rescue of NiPp1 phenotypes by human PPP1CC, a Pp1c ortholog with high similarity to all four *Drosophila* catalytic subunits (Figure 2G,H). Thus, these data agree with known overlapping/redundant roles for Pp1c during *Drosophila* development (Kirchner et al., 2007).

We edited the text to reflect these findings: “NiPp1 expression results in more severe phenotypes than RNAi knockdown, or loss, of individual Pp1c genes, at least with respect to migration and cluster cohesion, suggesting that Pp1c subunits have both distinct and overlapping functions. In particular, Pp1α-96A and Flw appear to function in border cell delamination, whereas all four subunits likely promote migration and cluster cohesion.”

2) In addition, the authors conclude that Pp1 globally coordinates the collective behaviors of border cells, from adhesion of cells within the cluster, to formation of migratory protrusions, formation of proper cell shapes, and efficient group motility. From the study it is unclear if these phenotypes are directly controlled by Pp1 or if some of them could be, for example, secondary to the loss of adhesion or the mis-localization of Myo-II. The authors mention in the Discussion section that Myo-II and cadherin-catenin complexes possess dynamic interactions that influence stability of cell-cell junctions, suggesting that the adhesion defect could be secondary to the defect in Myo-II distribution.It would be helpful to already mention and test such possible interactions in the result section. The authors should determine how F-actin, Myo-II and cadherin-catenin complexes are localized in the respective RNAi experiments.

This is an important point. As suggested, we determined if the phenotypes observed with Pp1-inhibition are secondarily due to decreases in the cadherin-catenin complex. We analyzed F-actin and active Myo-II (p-Sqh) in *E-Cad-*RNAi and *β-Cat-*RNAi border cells (Figure 6—figure supplement 2). Additionally, we analyzed the dynamics of migrating border cells when *α-Catenin* was knocked down by RNAi (Figure 5—figure supplement 1F-J). Moreover, Montell and colleagues recently examined Sqh protein dynamics in live *E-cad*-RNAi border cells (Mishra et al., 2019).

First, Pp1-inhibition resulted in very high F-actin with elevated peak fluorescence signals at the cell cortex of each individual border cell (Figure 6E,E’,G). In contrast, *cadherin-catenin*-RNAi resulted in uniform (and possibly lower) F-actin levels across the entire cluster (Figure 6—figure supplement 2B-C’,F,G). Second, the effects on Sqh/Myo-II were also different between NiPp1 and *E-Cad*/*β-Cat-*RNAi. Unlike control, p-Sqh was no longer enriched at the cluster periphery in *E-Cad-*RNAi or *β-Cat-*RNAi border cells (Figure 6—figure supplement 2I-K). This is in contrast to NiPp1 border cells, in which p-Sqh is localized both at the cluster periphery and between border cells (Figure 7A-C). Please note that we were unable to obtain confocal images for p-Sqh in control RNAi, *E-Cad-*RNAi, and *β-Cat-*RNAi, because the K-State Vet-Med Confocal Core shut down due to COVID-19. The images in Figure 6—figure supplement 2I-K were instead taken on our widefield fluorescent microscope. Because we did not include a membrane marker in these experiments, we did not quantitate the ratio of p-Sqh at different cell-cell contacts within the cluster. Consistent with our results with p-Sqh in *E-Cad-*RNAi and *β-Cat-*RNAi, however, Mishra et al., 2019, similarly observed overall reduction in cortical Sqh-mCherry protein distribution and levels in live *E-Cad-*RNAi border cells compared to control.

As suggested, we analyzed E-cadherin and β-Catenin in *sqh*-RNAi (Figure 6—figure supplement 3), allowing us to more directly determine how much Myo-II regulates the cadherin-catenin complex in border cells. We found little difference in E-cadherin levels at BC-BC cell contacts in *sqh*-RNAi compared to control (Figure 6—figure supplement 3A-B’’,E), although there was slightly elevated β-Catenin in *sqh-*RNAi border cell clusters (Figure 6—figure supplement 3C-D’’,F). We also analyzed F-actin levels and distribution in *sqh*-RNAi (Figure 6—figure supplement 2D,D’,H). We found that F-actin was present but no longer enriched at the periphery of *sqh-*RNAi border cell clusters (Figure 6—figure supplement 2D,D’,H). Thus, data from *sqh-*RNAi suggest that Myo-II is not a major regulator of the cadherin-catenin complex in border cells. Further, Myo-II may normally elevate F-actin levels at the cluster periphery.

We have added a section to the Results to describe these new findings, “Interplay between cadherin-catenin adhesion and actomyosin dynamics”. Our results support a more direct role for Pp1 activity in controlling collective versus single cell dynamics of actomyosin and cadherin-catenin in border cells. We favor a model in which there are distinct Pp1 phosphatase complexes that regulate Myo-II versus the cadherin-catenin complex as described in the Discussion.

3) The results pertaining to p-Sqh/Myosin and Mbs-RNAi, in particular, are less convincing and need to be strengthened (Figure 7) as the phenotypes were not very clear (e.g. Video 14).

We repeated the p-Sqh staining, using the same fixation and antibody staining protocols we used in our prior publication (Majumder et al., 2012), which resulted in consistent staining (see also Materials and methods). Moreover, we drove expression of NiPp1 using the early GAL4, *c306-*GAL4, along with a membrane-GFP (PLCδ-PH-EGFP). This allowed us to better quantify the relative ratio of p-Sqh at cell membrane contacts between border cells (“BC-BC”) and at contacts between border cells and nurse cells (“BC-NC”). We imaged many of these samples using the Zeiss Airyscan near-super resolution microscope, while others were imaged on our standard widefield fluorescent microscope. Please note that we were unable to access all of the Airyscan data after the K-State Confocal Facility closed due to COVID-19, but data were consistent no matter the acquisition method.

Border cells inhibited for Pp1 (NiPp1-expressing) exhibited high levels of p-Sqh at BC-BC contacts rather than just at the cluster periphery (BC-NC; Figure 7A-C; Videos 14-17 for Sqh-GFP). We measured the relative ratio of BC-NC p-Sqh to BC-BC p-Sqh. In control clusters, this ratio was ~1.10, indicating higher p-Sqh at BC-NC junctions. In NiPp1 clusters, the ratio was ~0.95, indicating an increase in p-Sqh at BC-BC junctions along with enriched p-Sqh at BC-NC junctions.

We edited the text: “Control border cells exhibited p-Sqh signal primarily at the cluster periphery (“BC-NC” contacts; Figure 7A-A’’). This pattern of p-Sqh closely resembles the pattern of Sqh-GFP in live wild-type border cells (Figure 6H-H””’) (Aranjuez et al., 2016; Majumder et al., 2012; Plutoni et al., 2019; Zeledon et al., 2019). NiPp1 border cells, however, had high levels of p-Sqh distributed throughout the cluster including at internal BC-BC contacts (Figure 7B-B’’), similar to Sqh-GFP in live NiPp1 border cells (Figure 6I-I””’). We measured the relative ratio of p-Sqh fluorescence intensity at BC-NC contacts versus BC-BC contacts in control and NiPp1 border cell clusters (Figure 7C). Control border cells had a higher p-Sqh ratio than NiPp1, indicating more p-Sqh signal at BC-NC contacts. These data support the idea that Pp1 inhibition elevates Myo-II activation within single border cells.”

For *Mbs* RNAi, we now show two examples of typical phenotypes in Figure 7D-F. In some cases, we observed splitting of the cluster, with one group staying at the anterior and the other group completing migration to the oocyte (Figure 7E), similar to what we observed with *flw* mosaic mutant clusters (*see* Figure 3H-H’’ and Figure 3—figure supplement 1B-B’’). In other cases, we observed both splitting along the migration path and migration defects (Figure 7F). We agree that *Mbs* RNAi migration and cohesion defects are not as penetrant as those observed with NiPp1 expression or *Pp1c* RNAi (Figure 7G,H compared to Figures 1I,J and 3E,F).

The milder phenotypes caused by *Mbs* RNAi could be due to incomplete knockdown by the *Mbs* RNAi transgene, even though Mbs protein levels were reduced (Figure 7—figure supplement 1K-L’’). Alternatively, the Pp1c/Mbs myosin phosphatase complex may be only one of multiple Pp1 complexes required in border cells. Although we believe that a major target of Pp1 activity is Myo-II, we also think it very likely that multiple Pp1 complexes regulate F-actin and adhesion and possibly other to-be-identified phosphorylated substrates in border cells (*see* Discussion).

We revised the text to reflect these results: “The phenotypes observed with Mbs-RNAi are milder than those observed with Pp1-inhibition (either NiPp1 or Pp1c-RNAi; compare to Figures 1I and 3E). This could be due to incomplete knockdown by Mbs-RNAi, although we observed significant decreases in the levels of endogenous Mbs (Figure 7—figure supplement 1K-L”). Alternatively, myosin phosphatase, through a complex of Mbs/Pp1c, could be one of multiple Pp1 complexes required for border cell cluster migration and cohesion (see Discussion). Nonetheless, these findings indicate that myosin phosphatase, a specific Pp1 complex, helps promote the normal cell morphology and collective cohesion of border cells, in addition to facilitating the successful migration of the border cells.”

Likewise, Video 16 does not convincingly support the conclusion that sqh:GFP is more uniform, “especially at contacts between some border cells”.

Because the previous videos of Sqh-GFP were acquired on a single focal plane to minimize phototoxicity while still having high resolution, we acknowledge that the videos may not have been clear. Therefore, we obtained new videos using full *z-*stacks. We reduced the laser intensity, used a faster frame rate, and further opened the pinhole, all of which achieved a better signal-to-noise ratio and faster acquisition time while still minimizing phototoxicity (Videos 14-17). We imaged for 20 min. Additionally, to more clearly show localization of Sqh-GFP in the cluster, in the new videos we included full 3D *z-*stack reconstructions of the border cell cluster at the start and end of imaging, along with the time-lapse video showing a single focal plane.

The new videos more clearly show the differences in Sqh-GFP localization in control versus Pp1-inhibited border cells. In NiPp1 border cells, Sqh-GFP no longer localizes to the cluster periphery, but is now found around each border cell. However, rather than being “more uniform” around each NiPp1-expressing border cell as stated in the previous version of the manuscript, we now find that Sqh-GFP is enriched in highly dynamic foci at the cell membranes of individual NiPp1-expressing border cells (Videos 15 and 17). These dynamic Sqh-GFP foci resemble those of control clusters (Videos 14 and 16), and are consistent with previous studies (Aranjuez et al., 2016; Combedazou et al., 2017; Majumder et al., 2012; Mishra et al., 2019). These results further support the idea that NiPp1-expressing border cells function more as single cells rather than as a collective.

These data are now shown in Video 14 (control early migration), Video 15 (NiPp1 early migration), Video 16 (control mid-migration), Video 17 (NiPp1 mid-migration), Figure 6H-I’’’’’, and Figure 6—figure supplement 1.

We edited the text: “In NiPp1 border cells, however, Sqh-GFP was now present at cortical cell membranes in dynamic foci surrounding each border cell (or sub-cluster) rather than at the entire cluster periphery, both during early migration (Figure 6—figure supplement 1B-B””’; Video 15) and at mid-migration stages (Figure 6I-I””’; Video 17). Thus, inhibition of Pp1 converts collectively polarized F-actin and Myo-II to that characteristic of single migrating cells. As a result, individual border cells now have enriched and dynamic actomyosin localization consistent with elevated cortical contractility in single cells rather than at the collective level.”

“Pp1c border cells followed the normal migratory path down the center of the egg chamber between nurse cells, even if when cells broke off…” This does not seem to be the case in Video 7.

We occasionally observed off-track migration, such as in this Video (now Video 8) and in some fixed samples (e.g. Figure 3C). However, the majority of Pp1-inhibited border cells stayed on the normal pathway, or very close to the normal pathway down the center of the egg chamber (Videos 2-4, 7, 9, 10). This is in contrast to mutants that disrupt directional migration, such as loss of the guidance receptors PVR and EGFR (Duchek et al., 2001), ectopic guidance ligands (McDonald et al., 2003), or loss of the cadherin-catenin complex (e.g. Figure 4B,D; Cai et al., 2014; although note that in Video 11, this example of *α-Cat-RNAi* border cells appears to be “on-track” during the imaging).

We have edited the text to reflect this: “The majority of NiPp1 and Pp1c RNAi border cells followed the normal migratory pathway down the center of the egg chamber between nurse cells, even when cells broke off from the main cluster (Figures 1H, L-L” and 3B-D; Videos 2-5, 7-10).”

Please, also indicate how many videos were obtained for each experiment.

The numbers of videos for each experiment/genotype are now stated in all of the figure legends and/or video legends.

4) Furthermore, while this study does a nice job of putting a lot of data into context, some results need to be more clearly explained in the context of the field, to clarify new results from those that have been previously documented- particularly with respect to the previous results of the Mbs subunit of PP1 (Aranjuez et al., 2016)

The focus of our previous studies of myosin phosphatase (Mbs) was on delamination, migration, and cluster shape (Aranjuez et al., 2016; Majumder et al., 2012). Dissociation of clusters was observed with *Mbs-*RNAi, similar to NiPp1, but was not quantified at the time (Aranjuez et al., 2016). Because the focus of the Aranjuez et al., 2016, study was on overall cluster morphology, rather than cohesion, we had not included these data.

We state this in the text: “Previously, we found that Mbs was required for border cell cluster delamination from the epithelium and cell shape (Aranjuez et al., 2016; Majumder et al., 2012), although cluster cohesion had not been explicitly assessed.”

In the current study we re-assessed the phenotypes caused by *Mbs-*RNAi, as shown in Figure 7D-H. While *Mbs-*RNAi only mildly disrupts migration, we observed significant dissociation of Mbs-deficient border cell clusters. While our manuscript was in preparation, Montell and colleagues published a study showing that expression of an activated form of Sqh (Sqh-^E20E21^) causes border cells to round up and dissociate from the cluster (Mishra et al., 2019). However, because Mishra et al., 2019, did not report the penetrance of these phenotypes, we cannot completely compare their results to what we observe with Pp1-inhibition. Likewise, we and others previously found that constitutively-activated Rho (Rho^V14^) disrupts border cell shape similar to Pp1-inhibition, with rounding of border cells and high F-actin and p-Sqh at individual border cell membrane cortices (Aranjuez et al., 2016; Combedazou et al., 2017; Mishra et al., 2019). Nonetheless, these data support our findings, as loss of myosin phosphatase (*Mbs-*RNAi) results in elevated p-Sqh (activated Myo-II) in border cells (Majumder et al., 2012). Thus, at least some of the phenotypes caused by Pp1-inhibition are due to loss of myosin phosphatase and high p-Sqh (activated Myo-II) around individual border cells (Figure 7A-C).

We have added this information to the Discussion: “Myosin phosphatase-depleted border cells, which have elevated phosphorylated Sqh and thus active Myo-II (Majumder et al., 2012), are highly contractile, round up, and fall off the cluster. In support of this idea, overexpression of a phosphorylation mutant form of Sqh (Sqh^E20E21^), which mimics activated Myo-II, caused border cells to similarly round up and separate from the cluster (Mishra et al., 2019).”

and if such drastic loss of border cell cohesion has been previously observed after any other experimental manipulations.

Such drastic loss of border cell cohesion has only been observed when E-cad was knocked down solely in polar cells, where it was reported that ~70% of clusters with *E-cad-*RNAi in polar cells are split (Cai et al., 2014). In the published literature, most cluster cohesion defects appear to be milder, with only a few border cells falling off the cluster or the cluster spreading out but not fully separating (Cai et al., 2014; Llense and Martín-Blanco, 2008; Melani et al., 2008; Raza et al., 2019). Interestingly, in each of these cases, adhesion is disrupted by loss of the respective gene. For example, the work by Raza et al., 2019, identified genes that co-evolve with E-cadherin and whose knockdown regulates cell-cell adhesion. The most severe phenotypes observed by RNAi of these genes was ~35% cluster dissociation (*Raskol-*RNAi), although the number of dissociated sub-clusters or “parts” was not reported (Raza et al., 2019). Knockdown of *Raskol* decreased levels of E-cadherin at BC-BC contacts, likely accounting for the cohesion defects. Our studies on Pp1 function found ~90% dissociated of NiPp1 clusters, with ~40% of egg chambers displaying more than 3 border cell parts. Inhibition of Pp1 thus appears to be more severe than those reported in previous studies, possibly due to Pp1 having multiple downstream targets.

To provide better context to our results, we now discuss similarities and differences between our results and previously published studies in the Discussion, paragraph five.